# Enhancing Sharpness-Aware Optimization Through Variance Suppression

**Bingcong Li**        **Georgios B. Giannakis**

University of Minnesota - Twin Cities
Minneapolis, MN, USA
{lixx5599, georgios}@umn.edu

## Abstract

Sharpness-aware minimization (SAM) has well documented merits in enhancing generalization of deep neural networks, even without sizable data augmentation. Embracing the geometry of the loss function, where neighborhoods of 'flat minima' heighten generalization ability, SAM seeks 'flat valleys' by minimizing the maximum loss caused by an *adversary* perturbing parameters within the neighborhood. Although critical to account for sharpness of the loss function, such an '*over-friendly* adversary' can curtail the outmost level of generalization. The novel approach of this contribution fosters stabilization of adversaries through *variance suppression* (VaSSO) to avoid such friendliness. VaSSO's *provable* stability safeguards its numerical improvement over SAM in model-agnostic tasks, including image classification and machine translation. In addition, experiments confirm that VaSSO endows SAM with robustness against high levels of label noise. Code is available at https://github.com/BingcongLi/VaSSO.

## 1   Introduction

Despite deep neural networks (DNNs) have advanced the concept of "learning from data," and markedly improved performance across several applications in vision and language (Devlin et al., 2018; Tom et al., 2020), their overparametrized nature renders the tendency to overfit on training data (Zhang et al., 2021). This has led to concerns in generalization, which is a practically underscored perspective yet typically suffers from a gap relative to the training performance.

Improving generalizability is challenging. Common approaches include (model) regularization and data augmentation (Srivastava et al., 2014). While it is the default choice to integrate regularization such as weight decay and dropout into training, these methods are often insufficient for DNNs especially when coping with complicated network architectures (Chen et al., 2022). Another line of effort resorts to suitable optimization schemes attempting to find a generalizable local minimum. For example, SGD is more preferable than Adam on certain overparameterized problems since it converges to maximum margin solutions (Wilson et al., 2017). Decoupling weight decay from Adam also empirically facilitates generalizability (Loshchilov and Hutter, 2017). Unfortunately, the underlying mechanism remains unclear, and whether the generalization merits carry over to other intricate learning tasks calls for additional theoretical elaboration.

Our main focus, sharpness aware minimization/optimization (SAM), is a highly compelling optimization approach that facilitates state-of-the-art generalizability by exploiting sharpness of loss landscape (Foret et al., 2021; Chen et al., 2022). A high-level interpretation of sharpness is how violently the loss fluctuates within a neighborhood. It has been shown through large-scale empirical studies that sharpness-based measures highly correlate with generalization (Jiang et al., 2019). Several works

37th Conference on Neural Information Processing Systems (NeurIPS 2023).

have successfully explored sharpness for generalization advances. For example, Keskar et al. (2016) suggests that the batchsize of SGD impresses solution flatness. Entropy SGD leverages local entropy in search of a flat valley (Chaudhari et al., 2017). Different from prior works, SAM induces flatness by explicitly minimizing the *adversarially* perturbed loss, defined as the maximum loss of a neighboring area. Thanks to such a formulation, SAM has elevated generalization merits among various tasks in vision and language domains (Chen et al., 2022; Zhang et al., 2022). The mechanism fertilizing SAM's success is also theoretically investigated based on arguments of implicit regularization; see e.g., (Andriushchenko and Flammarion, 2022; Wen et al., 2023; Bartlett et al., 2022).

The adversary perturbation, or *adversary* for short, is central to SAM's heightened generalization because it effectively measures sharpness through the loss difference with original model (Foret et al., 2021; Zhuang et al., 2022; Kim et al., 2022). In practice however, this awareness on sharpness is undermined by what we termed *friendly adversary*. Confined by the stochastic linearization for computational efficiency, SAM's adversary only captures the sharpness for a particular minibatch of data, and can become a friend on other data samples. Because the global sharpness is not approached accurately, the friendly adversary precludes SAM from attaining its utmost generalizability. The present work advocates variance suppressed sharpness aware optimization (VaSSO[1]) to alleviate 'friendliness' by stabilizing adversaries. With its *provable* stabilized adversary, VaSSO showcases favorable numerical performance on various deep learning tasks.

All in all, our contribution is summarized as follows.

- ❖ We find that the friendly adversary discourages generalizability of SAM. This challenge is catastrophic in our experiments – it can completely wipe out the generalization merits.

- ❖ A novel approach, VaSSO, is proposed to tackle this issue. VaSSO is equipped with what we termed *variance suppression* to streamline a principled means for stabilizing adversaries. The theoretically guaranteed stability promotes refined global sharpness estimates, thereby alleviating the issue of friendly adversary.

- ❖ A side result is tighter convergence analyses for VaSSO and SAM that i) remove the bounded gradient assumption; and ii) deliver a more flexible choice for hyperparameters.

- ❖ Numerical experiments confirm the merits of stabilized adversary in VaSSO. It is demonstrated on image classification and neural machine translation tasks that VaSSO is capable of i) improving generalizability over SAM model-agnostically; and ii) nontrivially robustifying neural networks under the appearance of large label noise.

**Notation**. Bold lowercase (capital) letters denote column vectors (matrices); $\|\mathbf{x}\|$ stands for $\ell_2$ norm of vector $\mathbf{x}$; and $\langle \mathbf{x}, \mathbf{y} \rangle$ is the inner product of $\mathbf{x}$ and $\mathbf{y}$. $\mathbb{S}_\rho(\mathbf{x})$ denotes the surface of a ball with radius $\rho$ centered at $\mathbf{x}$, i.e., $\mathbb{S}_\rho(\mathbf{x}) := \{\mathbf{x} + \rho\mathbf{u} \mid \|\mathbf{u}\| = 1\}$.

## 2   The known, the good, and the challenge of SAM

This section starts with a brief recap of SAM (i.e., the known), followed with refined analyses and positive results regarding its convergence (i.e., the good). Lastly, the *friendly adversary* issue is explained in detail and numerically illustrated.

### 2.1   The known

Targeting at a minimum in flat basin, SAM enforces small loss around the entire neighborhood in the parameter space (Foret et al., 2021). This idea is formalized by a minimax problem

$$\min_{\mathbf{x}} \max_{\|\boldsymbol{\epsilon}\| \leq \rho} f(\mathbf{x} + \boldsymbol{\epsilon}) \tag{1}$$

where $\rho$ is the radius of considered neighborhood, and the nonconvex objective is defined as $f(\mathbf{x}) := \mathbb{E}_{\mathcal{B}}[f_{\mathcal{B}}(\mathbf{x})]$. Here, $\mathbf{x}$ is the neural network parameter, and $\mathcal{B}$ is a random batch of data. The merits of such a formulation resides in its implicit sharpness measure $\max_{\|\boldsymbol{\epsilon}\| \leq \rho} f(\mathbf{x} + \boldsymbol{\epsilon}) - f(\mathbf{x})$, which effectively drives the optimization trajectory towards the desirable flat valley (Kim et al., 2022).

---

[1]Vasso coincides with the Greek nickname for Vasiliki.

---
**Algorithm 1** Generic form of SAM
---
1: **Initialize:** $\mathbf{x}_0, \rho$
2: **for** $t = 0, \ldots, T - 1$ **do**
3:     Sample a minibatch $\mathcal{B}_t$, and define stochastic gradient on $\mathcal{B}_t$ as $\mathbf{g}_t(\cdot)$
4:     Find $\boldsymbol{\epsilon}_t \in \mathbb{S}_\rho(\mathbf{0})$ via stochastic linearization; e.g., (4) for **VaSSO** or (3) for **SAM**
5:     Calculate stochastic gradient $\mathbf{g}_t(\mathbf{x}_t + \boldsymbol{\epsilon}_t)$
6:     Update model via $\mathbf{x}_{t+1} = \mathbf{x}_t - \eta\mathbf{g}_t(\mathbf{x}_t + \boldsymbol{\epsilon}_t)$
7: **end for**
8: **Return:** $\mathbf{x}_T$
---

The inner maximization of (1) has a natural interpretation as finding an *adversary*. Critical as it is, obtaining an adversary calls for *stochastic linearization* to alleviate computational concerns, i.e.,

$$\boldsymbol{\epsilon}_t = \arg\max_{\|\boldsymbol{\epsilon}\| \leq \rho} f(\mathbf{x}_t + \boldsymbol{\epsilon}) \overset{(a)}{\approx} \arg\max_{\|\boldsymbol{\epsilon}\| \leq \rho} f(\mathbf{x}_t) + \langle \nabla f(\mathbf{x}_t), \boldsymbol{\epsilon} \rangle \overset{(b)}{\approx} \arg\max_{\|\boldsymbol{\epsilon}\| \leq \rho} f(\mathbf{x}_t) + \langle \mathbf{g}_t(\mathbf{x}_t), \boldsymbol{\epsilon} \rangle \quad (2)$$

where linearization $(a)$ relies on the first order Taylor expansion of $f(\mathbf{x}_t + \boldsymbol{\epsilon})$. This is typically accurate given the choice of a small $\rho$. A stochastic gradient $\mathbf{g}_t(\mathbf{x}_t)$ then substitutes $\nabla f(\mathbf{x}_t)$ in $(b)$ to downgrade the computational burden of a full gradient. Catalyzed by the stochastic linearization in (2), it is possible to calculate SAM's adversary in closed-form

$$\boxed{\textbf{SAM:} \quad \boldsymbol{\epsilon}_t = \rho \frac{\mathbf{g}_t(\mathbf{x}_t)}{\|\mathbf{g}_t(\mathbf{x}_t)\|}.} \quad (3)$$

SAM then adopts the stochastic gradient of adversary $\mathbf{g}_t(\mathbf{x}_t + \boldsymbol{\epsilon}_t)$ to update $\mathbf{x}_t$ in a SGD fashion. A step-by-step implementation is summarized in Alg. 1, where the means to find an adversary in line 4 is presented in a generic form in order to unify the algorithmic framework with later sections.

## 2.2 The good

To provide a comprehensive understanding about SAM, this subsection focuses on Alg. 1, and establishes its convergence for (1). Some necessary assumptions are listed below, all of which are common for nonconvex stochastic optimization problems (Ghadimi and Lan, 2013; Bottou et al., 2016; Mi et al., 2022; Zhuang et al., 2022).

**Assumption 1** (lower bounded loss). $f(\mathbf{x})$ *is lower bounded, i.e.,* $f(\mathbf{x}) \geq f^*, \forall\mathbf{x}$.

**Assumption 2** (smoothness). *The stochastic gradient* $\mathbf{g}(\mathbf{x})$ *is L-Lipschitz, i.e.,* $\|\mathbf{g}(\mathbf{x}) - \mathbf{g}(\mathbf{y})\| \leq L\|\mathbf{x} - \mathbf{y}\|, \forall\mathbf{x}, \mathbf{y}$.

**Assumption 3** (bounded variance). *The stochastic gradient* $\mathbf{g}(\mathbf{x})$ *is unbiased with bounded variance, that is,* $\mathbb{E}[\mathbf{g}(\mathbf{x})|\mathbf{x}] = \nabla f(\mathbf{x})$ *and* $\mathbb{E}[\|\mathbf{g}(\mathbf{x}) - \nabla f(\mathbf{x})\|^2|\mathbf{x}] = \sigma^2$ *for some* $\sigma > 0$.

The constraint of (1) is never violated since $\|\boldsymbol{\epsilon}_t\| = \rho$ holds for each $t$; see line 4 in Alg. 1. Hence, the convergence of SAM pertains to the behavior of objective, where a tight result is given below.

**Theorem 1** (SAM convergence). *Suppose that Assumptions 1 – 3 hold. Let* $\eta_t \equiv \eta = \frac{\eta_0}{\sqrt{T}} \leq \frac{2}{3L}$, *and* $\rho = \frac{\rho_0}{\sqrt{T}}$. *Then with* $c_0 = 1 - \frac{3L\eta}{2}$ *(clearly* $0 < c_0 < 1$*), Alg. 1 guarantees that*

$$\frac{1}{T} \sum_{t=0}^{T-1} \mathbb{E}\big[\|\nabla f(\mathbf{x}_t)\|^2\big] \leq \mathcal{O}\left(\frac{\sigma^2}{\sqrt{T}}\right) \quad and \quad \frac{1}{T} \sum_{t=0}^{T-1} \mathbb{E}\big[\|\nabla f(\mathbf{x}_t + \boldsymbol{\epsilon}_t)\|^2\big] \leq \mathcal{O}\left(\frac{\sigma^2}{\sqrt{T}}\right).$$

The convergence rate of SAM is the same as SGD up to constant factors, where the detailed expression hidden under big $\mathcal{O}$ notation can be found in Appendix D. Our results eliminate the need for the bounded gradient assumption compared to existing analyses in (Mi et al., 2022; Zhuang et al., 2022). Moreover, Theorem 1 enables a much larger choice of $\rho = \mathcal{O}(T^{-1/2})$ relative to (Andriushchenko and Flammarion, 2022), where the latter only supports $\rho = \mathcal{O}(T^{-1/4})$.

A message from Theorem 1 is that *any* adversary satisfying $\boldsymbol{\epsilon}_t \in \mathbb{S}_\rho(\mathbf{0})$ ensures converge. Because the surface $\mathbb{S}_\rho(\mathbf{0})$ is a gigantic space, it challenges the plausible optimality of the adversary and poses a natural question – *is it possible to find a more powerful adversary for generalization advances?*

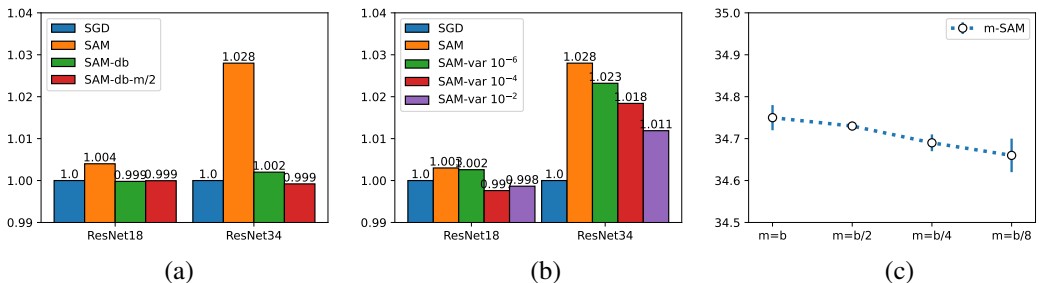

Figure 1: (a) A friendly adversary erases the generalization merits of SAM; (b) $m$-sharpness may *not* directly correlate with variance since noisy gradient degrades generalization; and (c) $m$-sharpness may not hold universally. Note that test accuracies in (a) and (b) are normalized to SGD.

## 2.3 The challenge: friendly adversary

**Adversary to one minibatch is a friend of others.** SAM's adversary is 'malicious' for minibatch $\mathcal{B}_t$ but not necessarily for other data because it only safeguards $f_{\mathcal{B}_t}(\mathbf{x}_t + \boldsymbol{\epsilon}_t) - f_{\mathcal{B}_t}(\mathbf{x}_t) \geq 0$ for a small $\rho$. In fact, it can be shown that $f_{\mathcal{B}}(\mathbf{x}_t + \boldsymbol{\epsilon}_t) - f_{\mathcal{B}}(\mathbf{x}_t) \leq 0$ whenever the stochastic gradients do not align well, i.e., $\langle \mathbf{g}_t(\mathbf{x}_t), \mathbf{g}_{\mathcal{B}}(\mathbf{x}_t) \rangle \leq 0$. Note that such misalignment is common because of the variance in massive training datasets. This issue is referred to as *friendly adversary*, and it implies that the adversary $\boldsymbol{\epsilon}_t$ cannot accurately depict the global sharpness of $\mathbf{x}_t$. Note that the 'friendly adversary' also has a more involved interpretation, that is, $\mathbf{g}_t(\mathbf{x}_t)$ falls outside the column space of Hessian at convergence; see more discussions after (Wen et al., 2023, Definition 4.3). This misalignment of higher order derivatives undermines the inductive bias of SAM, thereby worsening generalization.

To numerically visualize the catastrophic impact of the friendly adversary, we manually introduce one by replacing line 4 of Alg. 1 as $\tilde{\boldsymbol{\epsilon}}_t = \rho\tilde{\mathbf{g}}_t(\mathbf{x}_t)/\|\tilde{\mathbf{g}}_t(\mathbf{x}_t)\|$, where $\tilde{\mathbf{g}}_t$ denotes the gradient on $\tilde{\mathcal{B}}_t$, a randomly sampled batch of the same size as $\mathcal{B}_t$. This modified approach is denoted as SAM-db, and its performance for i) ResNet-18 on CIFAR10 and ii) ResNet-34 on CIFAR100[2] can be found in Fig. 1(a). Note that the test accuracy is normalized relative to SGD for the ease of visualization. It is evident that the friendly $\tilde{\boldsymbol{\epsilon}}_t$ in SAM-db almost erases the generalization benefits entirely.

**Source of friendly adversary.** The major cause to the friendly adversary attributes to the gradient variance, which equivalently translates to the lack of stability in SAM's stochastic linearization $(2b)$. An illustrative three dimensional example is shown in Fig. 2, where we plot the adversary $\boldsymbol{\epsilon}_t$ obtained from different $\mathbf{g}_t$ realization in $(2b)$. The minibatch gradient is simulated by adding Gaussian noise to the true gradient. When the signal to noise ration (SNR) is similar to a practical scenario (ResNet-18 on CIFAR10 shown in Fig. 2 (e)), it can be seen in Fig. 2 (c) and (d) that the adversaries *almost uniformly* spread over the norm ball, which strongly indicates the deficiency for sharpness evaluation.

**Friendly adversary in the lens of Frank Wolfe.** An additional evidence in supportive to SAM's friendly adversary resides in its connection to stochastic Frank Wolfe (SFW) that also heavily relies on stochastic linearization (Reddi et al., 2016). The stability of SFW is known to be vulnerable – its convergence cannot be guaranteed without a sufficient large batchsize. As thoroughly discussed in Appendix A, the means to obtain adversary in SAM is tantamount to one-step SFW with a *constant* batchsize. This symbolizes the possible instability of SAM's stochastic linearization.

## 2.4 A detailed look at friendly adversaries

The gradient variance is major cause to SAM's friendly adversary and unstable stochastic linearization, however this at first glance seems to conflict with an *empirical* note termed $m$-sharpness, stating that the benefit of SAM is clearer when $\boldsymbol{\epsilon}_t$ is found using subsampled $\mathcal{B}_t$ of size $m$ (i.e., larger variance).

Since $m$-sharpness highly hinges upon the loss curvature, it is unlikely to hold universally. For example, a transformer is trained on IWSLT-14 dataset, where the test performance (BLEU) decreases with smaller $m$ even if we have tuned $\rho$ carefully; see Fig. 1(c). On the theoretical side, an example is provided in (Andriushchenko and Flammarion, 2022, Sec. 3) to suggest that $m$-sharpness is not

---

[2]https://www.cs.toronto.edu/~kriz/cifar.html

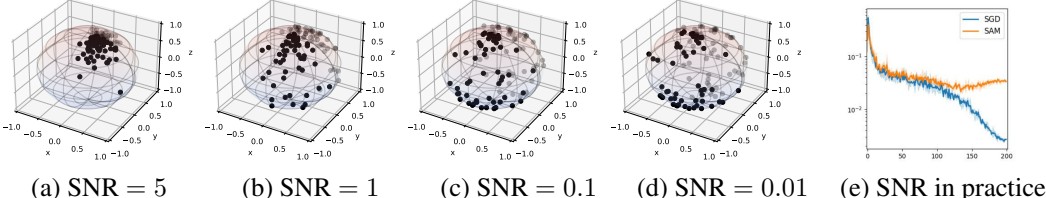

|     (a) SNR = 5     |     (b) SNR = 1     |     (c) SNR = 0.1     |     (d) SNR = 0.01     |     (e) SNR in practice     |

Figure 2: (a) - (d) SAM's adversaries spread over the surface; (e) SNR is in $[0.01, 0.1]$ when training a ResNet-18 on CIFAR10, where the SNR is calculated at the first iteration of every epoch.

necessarily related with sharpness or generalization. Moreover, there also exists specific choice for $m$ such that the $m$-sharpness formulation is ill-posed. We will expand on this in Appendix B.

Even in the regime where $m$-sharpness is empirically observed such as ResNet-18 on CIFAR10 and ResNet-34 on CIFAR100, we show through experiments that $m$-sharpness is *not* a consequence of gradient variance, thus not contradicting with the friendly adversary issue tackled in this work.

**Observation 1. Same variance, different generalization.** Let $m = 128$ and batchsize $b = 128$. Recall the SAM-db experiment in Fig. 1(a). If $m$-sharpness is a direct result of gradient variance, it is logical to expect SAM-db has comparable performance to SAM simply because their batchzises (hence variance) for finding adversary are the same. Unfortunately, SAM-db degrades accuracy. We further increase the variance of $\tilde{\mathbf{g}}_t(\mathbf{x}_t)$ by setting $m = 64$. The resultant algorithm is denoted as SAM-db-m/2. It does not catch with SAM and performs even worse than SAM-db. These experiments validate that variance/stability correlates with friendly adversary instead of $m$-sharpness.

**Observation 2. Enlarged variance degrades generalization.** We explicitly increase variance when finding adversary by adding Gaussian noise $\boldsymbol{\zeta}$ to $\mathbf{g}_t(\mathbf{x}_t)$, i.e., $\hat{\boldsymbol{\epsilon}}_t = \rho \frac{\mathbf{g}_t(\mathbf{x}_t) + \boldsymbol{\zeta}}{\|\mathbf{g}_t(\mathbf{x}_t) + \boldsymbol{\zeta}\|}$. After tuning the best $\rho$ to compensate the variance of $\boldsymbol{\zeta}$, the test performance is plotted in Fig. 1(b). It can be seen that the generalization merits clearly decrease with larger variance on both ResNet-18 and ResNet-34. This again illustrates that the plausible benefit of $m$-sharpness does not stem from increased variance.

In sum, observations 1 and 2 jointly suggest that gradient variance correlates with friendly adversary rather than $m$-sharpness, where understanding the latter is beyond the scope of current work.

## 3 Variance-supressed sharpness-aware optimization (VaSSO)

This section advocates variance suppression to handle the friendly adversary. We start with the design of VaSSO, then establish its stability. We also touch upon implementation and possible extensions.

### 3.1 Algorithm design and stability analysis

A straightforward attempt towards stability is to equip SAM's stochastic linearization with variance reduced gradients such as SVRG and SARAH (Johnson and Zhang, 2013; Nguyen et al., 2017; Li et al., 2019). However, the requirement to compute a full gradient every a few iterations is infeasible and hardly scales well for tasks such as training DNNs.

The proposed variance suppression (VaSSO) overcomes this computational burden through a novel yet simple stochastic linearization. For a prescribed $\theta \in (0, 1)$, VaSSO is summarized below

$$\textbf{VaSSO:} \quad \mathbf{d}_t = (1 - \theta)\mathbf{d}_{t-1} + \theta\mathbf{g}_t(\mathbf{x}_t) \tag{4a}$$

$$\boldsymbol{\epsilon}_t = \underset{\|\boldsymbol{\epsilon}\| \leq \rho}{\arg\max} f(\mathbf{x}_t) + \langle \mathbf{d}_t, \boldsymbol{\epsilon} \rangle = \rho \frac{\mathbf{d}_t}{\|\mathbf{d}_t\|}. \tag{4b}$$

Compared with (2) of SAM, the key difference is that VaSSO relies on slope $\mathbf{d}_t$ for a more stable stochastic linearization as shown in (4b). The slope $\mathbf{d}_t$ is an exponentially moving average (EMA) of $\{\mathbf{g}_t(\mathbf{x}_t)\}_t$ such that the change over consecutive iterations is smoothed. Noticing that $\boldsymbol{\epsilon}_t$ and $\mathbf{d}_t$ share the same direction, the relatively smoothed $\{\mathbf{d}_t\}_t$ thus imply the stability of $\{\boldsymbol{\epsilon}_t\}_t$ in VaSSO. Moreover, as $\mathbf{d}_t$ processes information of different minibatch data, the global sharpness can be captured in a principled manner to alleviate the friendly adversary challenge.

To theoretically characterize the effectiveness of VaSSO, our first result considers $\mathbf{d}_t$ as a qualified strategy to estimate $\nabla f(\mathbf{x}_t)$, and delves into its mean square error (MSE).

**Theorem 2** (Variance suppression). *Suppose that Assumptions 1 – 3 hold. Let Alg. 1 equip with i) $\boldsymbol{\epsilon}_t$ obtained by (4) with $\theta \in (0,1)$; and, ii) $\eta_t$ and $\rho$ selected the same as Theorem 1. VaSSO guarantees that the MSE of $\mathbf{d}_t$ is bounded by*

$$\mathbb{E}\big[\|\mathbf{d}_t - \nabla f(\mathbf{x}_t)\|^2\big] \leq \theta\sigma^2 + \mathcal{O}\left(\frac{(1-\theta)^2\sigma^2}{\theta^2\sqrt{T}}\right). \tag{5}$$

Because SAM's gradient estimate has a looser bound on MSE (or variance), that is, $\mathbb{E}[\|\mathbf{g}_t - \nabla f(\mathbf{x}_t)\|^2] \leq \sigma^2$, the shrunk MSE in Theorem 2 justifies the name of variance suppression.

Next, we quantify the stability invoked with the suppressed variance. It is convenient to start with necessary notation. Define the *quality* of a stochastic linearization at $\mathbf{x}_t$ with slope $\mathbf{v}$ as $\mathcal{L}_t(\mathbf{v}) := \max_{\|\boldsymbol{\epsilon}\| \leq \rho} f(\mathbf{x}_t) + \langle \mathbf{v}, \boldsymbol{\epsilon} \rangle$. For example, $\mathcal{L}_t(\mathbf{d}_t)$ and $\mathcal{L}_t\big(\mathbf{g}_t(\mathbf{x}_t)\big)$ are quality of VaSSO and SAM, respectively. Another critical case of concern is $\mathcal{L}_t\big(\nabla f(\mathbf{x}_t)\big)$. It is shown in (Zhuang et al., 2022) that $\mathcal{L}_t\big(\nabla f(\mathbf{x}_t)\big) \approx \max_{\|\boldsymbol{\epsilon}\| \leq \rho} f(\mathbf{x}_t + \boldsymbol{\epsilon})$ given a small $\rho$. Moreover, $\mathcal{L}_t\big(\nabla f(\mathbf{x}_t)\big) - f(\mathbf{x}_t)$ is also an accurate approximation to the sharpness (Zhuang et al., 2022). These observations safeguard $\mathcal{L}_t\big(\nabla f(\mathbf{x}_t)\big)$ as the anchor when analyzing the stability of SAM and VaSSO.

**Definition 1** ($\delta$-stability). *A stochastic linearization with slope $\mathbf{v}$ is said to be $\delta$-stable if its quality satisfies $\mathbb{E}\big[|\mathcal{L}_t(\mathbf{v}) - \mathcal{L}_t(\nabla f(\mathbf{x}_t))|\big] \leq \delta$.*

A larger $\delta$ implies a more friendly adversary, hence is less preferable. We are now well-prepared for our main results on adversary's stability.

**Theorem 3** (Adversaries of VaSSO is more stable than SAM.). *Suppose that Assumptions 1 – 3 hold. Under the same hyperparameter choices as Theorem 2, the stochastic linearization is $\big[\sqrt{\theta}\rho\sigma + \mathcal{O}(\frac{\rho\sigma}{\theta T^{1/4}})\big]$-stable for VaSSO, while $\rho\sigma$-stable in SAM.*

Theorem 3 demonstrates that VaSSO alleviates the friendly adversary problem by promoting stability. Qualitatively, VaSSO is roughly $\sqrt{\theta} \in (0,1)$ times more stable relative to SAM, since the term in big $\mathcal{O}$ notation is negligible given a sufficiently large $T$. Theorem 3 also guides the choice of $\theta$ – preferably small but not too small, otherwise the term in big $\mathcal{O}$ is inversely amplified.

## 3.2  Additional perspectives of VaSSO

Having discussed about the stability, this subsection proceeds with other aspects of VaSSO for a thorough characterization.

**Convergence.** Summarized in the following corollary, the convergence of VaSSO can be pursued as a direct consequence of Theorem 1. The reason is that $\boldsymbol{\epsilon}_t \in \mathbb{S}_\rho(\mathbf{0})$ is satisfied by (4).

**Corollary 1** (VaSSO convergence). *Suppose that Assumptions 1 – 3 hold. Choosing $\eta_t$ and $\rho$ the same as Theorem 1, then for any $\theta \in (0,1)$, VaSSO ensures that*

$$\frac{1}{T}\sum_{t=0}^{T-1}\mathbb{E}\big[\|\nabla f(\mathbf{x}_t)\|^2\big] \leq \mathcal{O}\left(\frac{\sigma^2}{\sqrt{T}}\right) \quad \text{and} \quad \frac{1}{T}\sum_{t=0}^{T-1}\mathbb{E}\big[\|\nabla f(\mathbf{x}_t + \boldsymbol{\epsilon}_t)\|^2\big] \leq \mathcal{O}\left(\frac{\sigma^2}{\sqrt{T}}\right).$$

**VaSSO better reflects sharpness around optimum.** Consider a near optimal region where $\|\nabla f(\mathbf{x}_t)\| \to 0$. Suppose that we are in a big data regime where $\mathbf{g}_t(\mathbf{x}_t) = \nabla f(\mathbf{x}_t) + \boldsymbol{\zeta}$ for some Gaussian random variable $\boldsymbol{\zeta}$. The covariance matrix of $\boldsymbol{\zeta}$ is assumed to be $\sigma^2\mathbf{I}$ for simplicity, but our discussion can be extended to more general scenarios using arguments from von Mises-Fisher statistics (Mardia and Jupp, 2000). SAM has difficulty to estimate the flatness in this case, since $\boldsymbol{\epsilon}_t \approx \rho\boldsymbol{\zeta}/\|\boldsymbol{\zeta}\|$ uniformly distributes over $\mathbb{S}_\rho(\mathbf{0})$ regardless of whether the neighboring region is sharp. On the other hand, VaSSO has $\boldsymbol{\epsilon}_t = \rho\mathbf{d}_t/\|\mathbf{d}_t\|$. Because $\{\mathbf{g}_\tau(\mathbf{x}_\tau)\}_\tau$ on sharper valley tend to have larger magnitude, their EMA $\mathbf{d}_t$ is helpful for distinguishing sharp with flat valleys.

**Memory efficient implementation.** Although at first glance VaSSO has to keep both $\mathbf{d}_t$ and $\boldsymbol{\epsilon}_t$ in memory, it can be implemented in a much more memory efficient manner. It is sufficient to store $\mathbf{d}_t$ together with a scaler $\|\mathbf{d}_t\|$ so that $\boldsymbol{\epsilon}_t$ can be recovered on demand through normalization; see (4b). Hence, VaSSO has the same memory consumption as SAM.

| CIFAR10 | SGD | SAM | ASAM | FisherSAM | VaSSO |
|---|---|---|---|---|---|
| **VGG-11-BN** | $93.20_{\pm0.05}$ | $93.82_{\pm0.05}$ | $93.47_{\pm0.04}$ | $93.60_{\pm0.09}$ | $\mathbf{94.10}_{\pm0.07}$ |
| **ResNet-18** | $96.25_{\pm0.06}$ | $96.58_{\pm0.10}$ | $96.33_{\pm0.09}$ | $96.72_{\pm0.03}$ | $\mathbf{96.77}_{\pm0.09}$ |
| **WRN-28-10** | $97.08_{\pm0.16}$ | $97.32_{\pm0.11}$ | $97.15_{\pm0.05}$ | $97.46_{\pm0.18}$ | $\mathbf{97.54}_{\pm0.12}$ |
| **PyramidNet-110** | $97.39_{\pm0.09}$ | $97.85_{\pm0.14}$ | $97.56_{\pm0.11}$ | $97.84_{\pm0.18}$ | $\mathbf{97.93}_{\pm0.08}$ |

Table 1: Test accuracy (%) of VaSSO on various neural networks trained on CIFAR10.

**Extensions.** VaSSO has the potential to boost the performance of other SAM family approaches by stabilizing their stochastic linearization through variance suppression. For example, adaptive SAM methods (Kwon et al., 2021; Kim et al., 2022) ensure scale invariance for SAM, and GSAM (Zhuang et al., 2022) jointly minimizes a surrogated gap with (1). Nevertheless, these SAM variants leverage stochastic linearization in (2). It is thus envisioned that VaSSO can also alleviate the possible friendly adversary issues therein. Confined by computational resources, we only integrate VaSSO with GSAM in our experiments, and additional evaluation has been added into our research agenda.

## 4 Numerical tests

To support our theoretical findings and validate the powerfulness of variance suppression, this section assesses generalization performance of VaSSO via various learning tasks across vision and language domains. All experiments are run on NVIDIA V100 GPUs.

### 4.1 Image classification

**Benchmarks.** Building on top of the selected base optimizer such as SGD and AdamW (Kingma and Ba, 2014; Loshchilov and Hutter, 2017), the test accuracy of VaSSO is compared with SAM and two adaptive approaches, ASAM and FisherSAM (Foret et al., 2021; Kwon et al., 2021; Kim et al., 2022).

**CIFAR10.** Neural networks including VGG-11, ResNet-18, WRN-28-10 and PyramidNet-110 are trained on CIFAR10. Standard implementation including random crop, random horizontal flip, normalization and cutout (Devries and Taylor, 2017) are leveraged for data augmentation. The first three models are trained for 200 epochs with a batchsize of 128, and PyramidNet-110 is trained for 300 epochs using batchsize 256. Cosine learning rate schedule is applied in all settings. The first three models use initial learning rate 0.05, and PyramidNet adopts 0.1. Weight decay is chosen as 0.001 for SAM, ASAM, FisherSAM and VaSSO following (Du et al., 2022a; Mi et al., 2022), but 0.0005 for SGD. We tune $\rho$ from $\{0.01, 0.05, 0.1, 0.2, 0.5\}$ for SAM and find that $\rho = 0.1$ gives the best results for ResNet and WRN, $\rho = 0.05$ and $\rho = 0.2$ suit best for and VGG and PyramidNet, respectively. ASAM and VaSSO adopt the same $\rho$ as SAM. FisherSAM uses the recommended $\rho = 0.1$ (Kim et al., 2022). For VaSSO, we tune $\theta = \{0.4, 0.9\}$ and report the best accuracy although VaSSO with both parameters outperforms SAM. We find that $\theta = 0.4$ works the best for ResNet-18 and WRN-28-10 while $\theta = 0.9$ achieves the best accuracy in other cases.

It is shown in Table 1 that VaSSO offers 0.2 to 0.3 accuracy improvement over SAM in all tested scenarios except for PyramidNet-110, where the improvement is about 0.1. These results illustrate that suppressed variance and the induced stabilized adversary are indeed beneficial for generalizability.

**CIFAR100.** The training setups on this dataset are the same as those on CIFAR10, except for the best choice for $\rho$ of SAM is 0.2. The numerical results are listed in Table 2. It can be seen that SAM has significant generalization gain over SGD, and this gain is further amplified by VaSSO. On all tested models, VaSSO improves the test accuracy of SAM by 0.2 to 0.3. These experiments once again corroborate the generalization merits of VaSSO as a blessing of the stabilized adversary.

**ImageNet.** Next, we investigate the performance of VaSSO on larger scale experiments by training ResNet-50 and ViT-S/32 on ImageNet (Deng et al., 2009). Implementation details are deferred to Appendix C. Note that the baseline optimizer is SGD for ResNet and AdamW for ViT. VaSSO is also integrated with GSAM (V+G) to demonstrate that the variance suppression also benefits other SAM type approaches (Zhuang et al., 2022). For ResNet-50, it can be observed that vanilla VaSSO

| CIFAR100 | SGD | SAM | ASAM | FisherSAM | VaSSO |
|---|---|---|---|---|---|
| **ResNet-18** | $77.90_{\pm 0.07}$ | $80.96_{\pm 0.12}$ | $79.91_{\pm 0.04}$ | $80.99_{\pm 0.13}$ | $\mathbf{81.30}_{\pm 0.13}$ |
| **WRN-28-10** | $81.71_{\pm 0.13}$ | $84.88_{\pm 0.10}$ | $83.54_{\pm 0.14}$ | $84.91_{\pm 0.07}$ | $\mathbf{85.06}_{\pm 0.05}$ |
| **PyramidNet-110** | $83.50_{\pm 0.12}$ | $85.60_{\pm 0.11}$ | $83.72_{\pm 0.09}$ | $85.55_{\pm 0.14}$ | $\mathbf{85.85}_{\pm 0.09}$ |

Table 2: Test accuracy (%) of VaSSO on various neural networks trained on CIFAR100.

| ImageNet | vanilla | SAM | ASAM | GSAM | VaSSO | V+G |
|---|---|---|---|---|---|---|
| **ResNet-50** | $76.62_{\pm 0.12}$ | $77.16_{\pm 0.14}$ | $77.10_{\pm 0.16}$ | $77.20_{\pm 0.13}$ | $\mathbf{77.42}_{\pm 0.13}$ | $\mathbf{77.48}_{\pm 0.04}$ |
| **ViT-S/32** | $68.12_{\pm 0.05}$ | $68.98_{\pm 0.08}$ | $68.74_{\pm 0.11}$ | $69.42_{\pm 0.18}$ | $\mathbf{69.54}_{\pm 0.15}$ | $\mathbf{69.61}_{\pm 0.11}$ |

Table 3: Test accuracy (%) of VaSSO on ImageNet, where V+G is short for VaSSO + GSAM.

outperforms other SAM variants, and offers a gain of $0.26$ over SAM. V+G showcases the best performance with a gain of $0.28$ on top of GSAM. VaSSO and V+G also exhibit the best test accuracy on ViT-S/32, where VaSSO improves SAM by $0.56$ and V+G outperforms GSAM by $0.19$. These numerical improvement demonstrates that stability of adversaries is indeed desirable.

## 4.2 Neural machine translation

Having demonstrated the benefits of a suppressed variance on vision tasks, we then test VaSSO on German to English translation using a Transformer (Vaswani et al., 2017) trained on IWSLT-14 dataset (Cettolo et al., 2014). The fairseq implementation is adopted. AdamW is chosen as base optimizer in SAM and VaSSO because of its improved performance over SGD. The learning rate of AdamW is initialized to $5 \times 10^{-4}$ and then follows an inverse square root schedule. For momentum, we choose $\beta_1 = 0.9$ and $\beta_2 = 0.98$. Label smoothing is also applied with a rate of $0.1$. Hyperparameter $\rho$ is tuned for SAM from $\{0.01, 0.05, 0.1, 0.2\}$, and $\rho = 0.1$ performs the best. The same $\rho$ is picked for ASAM and VaSSO as well.

The validation perplexity and test BLEU scores are shown in Table 4. It can be seen that both SAM and ASAM have better performance on validation perplexity and BLEU relative to AdamW. Although VaSSO with $\theta = 0.9$ has slightly higher validation perplexity, its BLEU score outperforms SAM and ASAM. VaSSO with $\theta = 0.4$ showcases the best generalization performance on this task, providing a $0.22$ improvement on BLEU score relative to AdamW. This aligns with Theorems 2 and 3, which suggest that a small $\theta$ is more beneficial to the stability of adversary.

## 4.3 Additional tests

Additional experiments are conducted to corroborate the merits of suppressed variance and stabilized adversary in VaSSO. In particular, this subsection evaluates several flatness related metrics after training a ResNet-18 on CIFAR10 for 200 epochs, utilizing the same hyperparameters as those in Section 4.1.

| | SGD | SAM | VaSSO |
|---|---|---|---|
| $\lambda_1$ | 82.52 | 26.40 | **23.32** |
| $\lambda_1/\lambda_5$ | 16.63 | 2.12 | **1.86** |

Table 5: Hessian spectrum of a ResNet-18 trained on CIFAR10.

**Hessian spectrum.** We first assess Hessian eigenvalues of a ResNet-18 trained with SAM and VaSSO. We focus on the largest eigenvalue $\lambda_1$ and the ratio of largest to the fifth largest eigenvalue $\lambda_1/\lambda_5$. These measurements are also adopted in (Foret et al., 2021; Jastrzebski et al., 2020) to reflect the flatness of the solution, where smaller numbers are more preferable. Because exact calculation for Hessian spectrum is too expensive provided the size of ResNet-18, we instead leverage Lanczos algorithm for approximation (Ghorbani et al., 2019). The results can be found in Table 5. It can be seen that SAM indeed converges to a much flatter solution compared with SGD, and VaSSO further improves upon SAM. This confirms that the friendly adversary issue is indeed alleviated by

|            | AdamW | SAM | ASAM | VaSSO $(\theta = 0.9)$ | VaSSO $(\theta = 0.4)$ |
|------------|-------|-----|------|------------------------|------------------------|
| val. ppl.  | $5.02_{\pm 0.03}$ | $5.00_{\pm 0.04}$ | $\mathbf{4.99}_{\pm 0.03}$ | $5.00_{\pm 0.03}$ | $\mathbf{4.99}_{\pm 0.03}$ |
| BLEU       | $34.66_{\pm 0.06}$ | $34.75_{\pm 0.04}$ | $34.76_{\pm 0.04}$ | $34.81_{\pm 0.04}$ | $\mathbf{34.88}_{\pm 0.03}$ |

Table 4: Performance of VaSSO for training a Transformer on IWSLT-14 dataset.

|                  | SAM | VaSSO $(\theta = 0.9)$ | VaSSO $(\theta = 0.4)$ | VaSSO $(\theta = 0.2)$ |
|------------------|-----|------------------------|------------------------|------------------------|
| **25% label noise** | $96.39_{\pm 0.12}$ | $96.36_{\pm 0.11}$ | $96.42_{\pm 0.12}$ | $\mathbf{96.48}_{\pm 0.09}$ |
| **50% label noise** | $93.93_{\pm 0.21}$ | $94.00_{\pm 0.24}$ | $94.63_{\pm 0.21}$ | $\mathbf{94.93}_{\pm 0.16}$ |
| **75% label noise** | $75.36_{\pm 0.42}$ | $77.40_{\pm 0.37}$ | $80.94_{\pm 0.40}$ | $\mathbf{85.02}_{\pm 0.39}$ |

Table 6: Test accuracy (%) of VaSSO on CIFAR10 under different levels of label noise.

the suppressed variance in VaSSO, which in turn boosts the generalizability of ResNet-18 as shown earlier in Section 4.1.

**Label noise.** It is known that SAM holds great potential to harness robustness to neural networks under the appearance of label noise in training data (Foret et al., 2021). As the training loss landscape is largely perturbed by the label noise, this is a setting where the suppressed variance and stabilized adversaries are expected to be advantageous. In our experiments, we measure the performance VaSSO in the scenarios where certain fraction of the training labels are randomly flipped. Considering $\theta = \{0.9, 0.4, 0.2\}$, the corresponding test accuracies are summarized in Table 6.

Our first observation is that VaSSO outperforms SAM at different levels of label noise. VaSSO elevates higher generalization improvement as the ratio of label noise grows. In the case of 75% label noise, VaSSO with $\theta = 0.4$ nontrivially outperforms SAM with an absolute improvement more than 5, while VaSSO with $\theta = 0.2$ markedly improves SAM by roughly 10. In all scenarios, $\theta = 0.2$ showcases the best performance and $\theta = 0.9$ exhibits the worst generalization when comparing among VaSSO. In addition, when fixing the choice to $\theta$, e.g., $\theta = 0.2$, it is found that VaSSO has larger absolute accuracy improvement over SAM under higher level of label noise. These observations coincide with Theorem 3, which predicts that VaSSO is suitable for settings with larger label noise due to enhanced stability especially when $\theta$ is chosen small (but not too small).

## 5 Other related works

This section discusses additional related work on generalizability of DNNs. The possibility of blending VaSSO with other approaches is also entailed to broaden the scope of this work.

**Sharpness and generalization.** Since the study of Keskar et al. (2016), the relation between sharpness and generalization has been intensively investigated. It is observed that sharpness is closely correlated with the ratio between learning rate and batchsize in SGD (Jastrzębski et al., 2017). Theoretical understandings on the generalization error using sharpness-related measures can be found in e.g., (Dziugaite and Roy, 2017; Neyshabur et al., 2017; Wang and Mao, 2022). These works justify the goal of seeking for a flatter valley to enhance generalizability. Targeting at a flatter minimum, approaches other than SAM are also developed. For example, Izmailov et al. (2018) proposes stochastic weight averaging for DNNs. Wu et al. (2020) studies a similar algorithm as SAM while putting more emphases on the robustness of adversarial training.

**Other SAM type approaches.** Besides the discussed ones such as GSAM and ASAM, (Zhao et al., 2022a) proposes a variant of SAM by penalizing the gradient norm based on the observation where sharper valley tends to have gradient with larger norm. Barrett and Dherin (2021) arrive at a similar conclusion by analyzing the gradient flow. Exploiting multiple (ascent) steps to find an adversary is systematically studied in (Kim et al., 2023). SAM has also been extended to tackle the challenges in

domain adaptation (Wang et al., 2023). However, these works overlook the friendly adversary issue, and the proposed VaSSO provides algorithmic possibilities for generalization benefits by stabilizing their adversaries. Since the desirable confluence with VaSSO can be intricate, we leave an in-depth investigation for future work.

**Limitation of VaSSO and possible solutions.** The drastically improved generalization of VaSSO comes at the cost of additional computation. Similar to SAM, VaSSO requires to backpropagate twice per iteration. Various works have tackled this issue and developed lightweight SAM. LookSAM computes the extra stochastic gradient once every a few iterations and reuses it in a fine-grained manner to approximate the additional gradient (Liu et al., 2022). ESAM obtains its adversary based on stochastic weight perturbation, and further saves computation by selecting a subset of the minibatch data for gradient computation (Du et al., 2022a). The computational burden of SAM can be compressed by switching between SAM and SGD following a predesigned schedule (Zhao et al., 2022b), or in an adaptive fashion (Jiang et al., 2023). SAF connects SAM with distillation for computational merits (Du et al., 2022b). It should be pointed out that most of these works follow the stochastic linearization of SAM, hence can also encounter the issue of friendly adversary. This opens the door of merging VaSSO with these approaches for generalization merits while respecting computational overhead simultaneously. This has been included in our research agenda.

## 6 Concluding remarks

This contribution demonstrates that stabilizing adversary through variance suppression consolidates the generalization merits of sharpness aware optimization. The proposed approach, VaSSO, provably facilitates stability over SAM. The theoretical merit of VaSSO reveals itself in numerical experiments, and catalyzes model-agnostic improvement over SAM among various vision and language tasks. Moreover, VaSSO nontrivially enhances model robustness against high levels of label noise. Our results corroborate VaSSO as a competitive alternative of SAM.

**Acknowledgement**

This research is supported by NSF grants 2128593, 2126052, 2212318, 2220292, and 2312547. The authors would also like to thank anonymous reviewers for their feedback.

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

# Supplementary Document for
## "Enhancing Sharpness-Aware Optimization Through Variance Suppression"

## A  Linking SAM adversary with stochastic Frank Wolfe

### A.1  Stochastic Frank Wolfe (SFW)

We first briefly review SFW. Consider the following nonconvex stochastic optimization

$$\max_{\mathbf{x} \in \mathcal{X}} h(\mathbf{x}) := \mathbb{E}_\xi \big[ h(\mathbf{x}, \xi) \big] \tag{6}$$

where $\mathcal{X}$ is a convex and compact constraint set. SFW for solving (6) is summarized below.

---
**Algorithm 2** SFW (Reddi et al., 2016)

---
1: **Initialize:** $\mathbf{x}_0 \in \mathcal{X}$
2: **for** $t = 0, 1, \ldots, T - 1$ **do**
3:     draw iid samples $\{\xi_t^b\}_{b=1}^{B_t}$
4:     let $\hat{\mathbf{g}}_t = \frac{1}{B_t} \sum_{b=1}^{B_t} \nabla h(\mathbf{x}_t, \xi_t^b)$
5:     $\mathbf{v}_{t+1} = \arg\max_{\mathbf{v} \in \mathcal{X}} \langle \hat{\mathbf{g}}_t, \mathbf{v} \rangle$
6:     $\mathbf{x}_{t+1} = (1 - \gamma_t)\mathbf{x}_t + \gamma_t \mathbf{v}_{t+1}$
7: **end for**

---

It has been shown in (Reddi et al., 2016, Theorem 2) that one has to use a sufficient large batch size $B_t = \mathcal{O}(T), \forall t$ to ensure convergence of SFW. This is because line 5 in Alg. 2 is extremely sensitive to gradient noise.

### A.2  The adversary of SAM

By choosing $h(\boldsymbol{\epsilon}) = f(\mathbf{x}_t + \boldsymbol{\epsilon})$ and $\mathcal{X} = \mathbb{S}_\rho(\mathbf{0})$, it is not hard to observe that 1-iteration SFW with $\gamma_0 = 1$ gives equivalent solution to the stochastic linearization in SAM; cf. (2) and (3). This link suggests that the SAM adversary also suffers from stability issues in the same way as SFW. Moreover, what amplifies this issue in SAM is the adoption of a constant batch size, which is typically small and far less than the $\mathcal{O}(T)$ requirement for SFW.

Our solution VaSSO takes inspiration from modified SFW approaches which leverage a constant batch size to ensure convergence; see e.g., (Mokhtari et al., 2020; Li et al., 2021). Even though, coping with SAM's instability is still challenging with two major obstacles. First, SAM uses *one-step* SFW, which internally breaks nice analytical structures. Moreover, the inner maximization (i.e., the objective function to the SFW) *varies every iteration* along with the updated $\mathbf{x}_t$.

### A.3  The three dimensional example in Fig. 2

Detailed implementation for Fig. 2 is listed below. We use $\nabla f(\mathbf{x}) = [0.2, -0.1, 0.6]$. The stochastic noise is $\boldsymbol{\xi} = [\xi_1, \xi_2, \xi_3]$, where $\xi_1, \xi_2, \xi_3$ are iid Gaussian random variables with variance scaling with 0.2, 1, 2, respectively. We scale the variance to change the SNR. We generate 100 adversaries by solving $\arg\max_{\|\boldsymbol{\epsilon}\| \leq \rho} \langle \nabla f(\mathbf{x}) + \boldsymbol{\xi}, \boldsymbol{\epsilon} \rangle$ for each choice of SNR. As shown in Fig. 2, the adversaries are unlikely to capture the sharpness information when the SNR is small, because they spread indistinguishably over the sphere.

## B  More on $m$-sharpness

$m$**-sharpness can be ill-posed.** Our reason for not studying $m$-sharpness directly is that its formulation (Andriushchenko and Flammarion, 2022, eq. (3)) may be ill-posed mathematically due to the lack of a clear definition on how the dataset $\mathcal{S}$ is partitioned. Consider the following example, where the same notation as (Andriushchenko and Flammarion, 2022) is adopted for convenience. Suppose that the loss function is $l_i(w) = a_i w^2 + b_i w$, where $(a_i, b_i)$ are data points and $w$ is the

parameter to be optimized. Let the dataset have $4$ samples, $(a_1 = 0, b_1 = 1)$; $(a_2 = 0, b_2 = -1)$; $(a_3 = -1, b_3 = 0)$; and, $(a_4 = 1, b_4 = 0)$. Consider 2-sharpness.

- If the data partition is {1,2} and {3,4}, the objective of 2-sharpness i.e., equation (3) in (Andriushchenko and Flammarion, 2022), becomes $\min_w \sum_{i=1}^2 \max_{||\delta|| < \rho} 0$.

- If the data partition is {1,3} and {2,4}, the objective is $\min_w \sum_{i=1}^2 \max_{||\delta|| < \rho} f_i(w, \delta)$, where $f_1$ is the loss on partition {1,3}, i.e., $f_1(w, \delta) = -(w + \delta)^2 + (w + \delta)$; and $f_2(w, \delta) = (w + \delta)^2 - (w + \delta)$ is the loss on partition {3,4}.

Clearly, the objective functions are different when the data partition varies. This makes the problem ill-posed – different manners of data partition lead to entirely different loss curvature. In practice, the data partition even vary with a frequency of an epoch due to the random shuffle.

# C   Details on numerical results

**CIFAR10 and CIFAR100.** For these small resolution datasets, we slightly change the first convolution layer of ResNet18 and WRN-28-10 to one with $3 \times 3$ kernel size, 1 stride and 1 padding following Mi et al. (2022). The results on SGD and SAM demonstrate that the accuracy is almost identical to the vanilla model.

**ResNet50 on ImageNet.** Due to the constraints on computational resources, we report the averaged results over 2 independent runs. For this dataset, we randomly resize and crop all images to a resolution of $224 \times 224$, and apply random horizontal flip, normalization during training. The batch size is chosen as $128$ with a cosine learning rate scheduling with an initial step size $0.05$. The momentum and weight decay of base optimizer, SGD, are set as $0.9$ and $10^{-4}$, respectively. We further tune $\rho$ from $\{0.05, 0.075, 0.1, 0.2\}$, and chooses $\rho = 0.075$ for SAM. VaSSO uses $\theta = 0.99$. VaSSO and ASAM adopt the same $\rho = 0.075$.

**ViT-S/32 on ImageNet.** We follow the implementation of (Du et al., 2022b), where we train the model for 300 epochs with a batch size of $4096$. The baseline optimizer is chosen as AdamW with weight decay $0.3$. SAM relies on $\rho = 0.05$. For the implementation of GSAM and V+G, we adopt the same implementation from (Zhuang et al., 2022).

# D   Missing proofs

Alg. 1 can be written as

$$\mathbf{x}_{t+\frac{1}{2}} = \mathbf{x}_t + \boldsymbol{\epsilon}_t \tag{7a}$$

$$\mathbf{x}_{t+1} = \mathbf{x}_t - \eta_t \mathbf{g}_t(\mathbf{x}_{t+\frac{1}{2}}) \tag{7b}$$

where $\|\boldsymbol{\epsilon}_t\| = \rho$. In SAM, we have $\boldsymbol{\epsilon}_t = \rho \frac{\mathbf{g}_t(\mathbf{x}_t)}{\|\mathbf{g}_t(\mathbf{x}_t)\|}$, and in VaSSO we have $\boldsymbol{\epsilon}_t = \rho \frac{\mathbf{d}_t}{\|\mathbf{d}_t\|}$.

## D.1   Useful lemmas

This subsection presents useful lemmas to support our main results.

**Lemma 1.** *Alg. 1 (or equivalently iteration (7)) ensures that*

$$\eta_t \mathbb{E}\big[\langle \nabla f(\mathbf{x}_t), \nabla f(\mathbf{x}_t) - \mathbf{g}_t(\mathbf{x}_{t+\frac{1}{2}}) \rangle \big] \le \frac{L \eta_t^2}{2} \mathbb{E}\big[\|\nabla f(\mathbf{x}_t)\|^2\big] + \frac{L \rho^2}{2}.$$

*Proof.* To start with, we have that

$$\langle \nabla f(\mathbf{x}_t), \nabla f(\mathbf{x}_t) - \mathbf{g}_t(\mathbf{x}_{t+\frac{1}{2}}) \rangle = \langle \nabla f(\mathbf{x}_t), \nabla f(\mathbf{x}_t) - \mathbf{g}_t(\mathbf{x}_t) + \mathbf{g}_t(\mathbf{x}_t) - \mathbf{g}_t(\mathbf{x}_{t+\frac{1}{2}}) \rangle. \tag{8}$$

Taking expectation conditioned on $\mathbf{x}_t$, we arrive at

$$
\mathbb{E}\big[\langle \nabla f(\mathbf{x}_t), \nabla f(\mathbf{x}_t) - \mathbf{g}_t(\mathbf{x}_{t+\frac{1}{2}})\rangle | \mathbf{x}_t\big]
$$
$$
= \mathbb{E}\big[\langle \nabla f(\mathbf{x}_t), \nabla f(\mathbf{x}_t) - \mathbf{g}_t(\mathbf{x}_t)\rangle | \mathbf{x}_t\big] + \mathbb{E}\big[\langle \nabla f(\mathbf{x}_t), \mathbf{g}_t(\mathbf{x}_t) - \mathbf{g}_t(\mathbf{x}_{t+\frac{1}{2}})\rangle | \mathbf{x}_t\big]
$$
$$
= \mathbb{E}\big[\langle \nabla f(\mathbf{x}_t), \mathbf{g}_t(\mathbf{x}_t) - \mathbf{g}_t(\mathbf{x}_{t+\frac{1}{2}})\rangle | \mathbf{x}_t\big]
$$
$$
\leq \mathbb{E}\big[\|\nabla f(\mathbf{x}_t)\| \cdot \|\mathbf{g}_t(\mathbf{x}_t) - \mathbf{g}_t(\mathbf{x}_{t+\frac{1}{2}})\| | \mathbf{x}_t\big]
$$
$$
\overset{(a)}{\leq} L\mathbb{E}\big[\|\nabla f(\mathbf{x}_t)\| \cdot \|\mathbf{x}_t - \mathbf{x}_{t+\frac{1}{2}}\| | \mathbf{x}_t\big]
$$
$$
\overset{(b)}{=} L\rho\|\nabla f(\mathbf{x}_t)\|
$$

where (a) is because of Assumption 2; and (b) is because $\mathbf{x}_t - \mathbf{x}_{t+\frac{1}{2}} = -\boldsymbol{\epsilon}_t$ and its norm is $\rho$. This inequality ensures that

$$
\eta_t \mathbb{E}\big[\langle \nabla f(\mathbf{x}_t), \nabla f(\mathbf{x}_t) - \mathbf{g}_t(\mathbf{x}_{t+\frac{1}{2}})\rangle | \mathbf{x}_t\big] \leq L\rho\eta_t\|\nabla f(\mathbf{x}_t)\| \leq \frac{L\eta_t^2\|\nabla f(\mathbf{x}_t)\|^2}{2} + \frac{L\rho^2}{2}
$$

where the last inequality is because $\rho\eta_t\|\nabla f(\mathbf{x}_t)\| \leq \frac{1}{2}\eta_t^2\|\nabla f(\mathbf{x}_t)\|^2 + \frac{1}{2}\rho^2$. Taking expectation w.r.t. $\mathbf{x}_t$ finishes the proof. $\qquad\square$

**Lemma 2.** *Alg. 1 (or equivalently iteration (7)) ensures that*

$$
\mathbb{E}\big[\|\mathbf{g}_t(\mathbf{x}_{t+\frac{1}{2}})\|^2\big] \leq 2L^2\rho^2 + 2\mathbb{E}\big[\|\nabla f(\mathbf{x}_t)\|^2\big] + 2\sigma^2.
$$

*Proof.* The proof starts with bounding $\|\mathbf{g}_t(\mathbf{x}_{t+\frac{1}{2}})\|$ as

$$
\|\mathbf{g}_t(\mathbf{x}_{t+\frac{1}{2}})\|^2 = \|\mathbf{g}_t(\mathbf{x}_{t+\frac{1}{2}}) - \mathbf{g}_t(\mathbf{x}_t) + \mathbf{g}_t(\mathbf{x}_t)\|^2
$$
$$
\leq 2\|\mathbf{g}_t(\mathbf{x}_{t+\frac{1}{2}}) - \mathbf{g}_t(\mathbf{x}_t)\|^2 + 2\|\mathbf{g}_t(\mathbf{x}_t)\|^2
$$
$$
\overset{(a)}{\leq} 2L^2\|\mathbf{x}_t - \mathbf{x}_{t+\frac{1}{2}}\|^2 + 2\|\mathbf{g}_t(\mathbf{x}_t)\|^2
$$
$$
\overset{(b)}{=} 2L^2\rho^2 + 2\|\mathbf{g}_t(\mathbf{x}_t) - \nabla f(\mathbf{x}_t) + \nabla f(\mathbf{x}_t)\|^2
$$

where (a) is the result of Assumption 2; and (b) is because $\mathbf{x}_t - \mathbf{x}_{t+\frac{1}{2}} = -\boldsymbol{\epsilon}_t$ and its norm is $\rho$.

Taking expectation conditioned on $\mathbf{x}_t$, we have

$$
\mathbb{E}\big[\|\mathbf{g}_t(\mathbf{x}_{t+\frac{1}{2}})\|^2 | \mathbf{x}_t\big] \leq 2L^2\rho^2 + 2\mathbb{E}\big[\|\mathbf{g}_t(\mathbf{x}_t) - \nabla f(\mathbf{x}_t) + \nabla f(\mathbf{x}_t)\|^2 | \mathbf{x}_t\big]
$$
$$
\leq 2L^2\rho^2 + 2\|\nabla f(\mathbf{x}_t)\|^2 + 2\sigma^2
$$

where the last inequality is because of Assumption 3. Taking expectation w.r.t. the randomness in $\mathbf{x}_t$ finishes the proof. $\qquad\square$

**Lemma 3.** *Let $A_{t+1} = \alpha A_t + \beta$ with some $\alpha \in (0, 1)$, then we have*

$$
A_{t+1} \leq \alpha^{t+1}A_0 + \frac{\beta}{1 - \alpha}.
$$

*Proof.* The proof can be completed by simply unrolling $A_{t+1}$ and using the fact $1 + \alpha + \alpha^2 + \ldots + \alpha^t \leq \frac{1}{1-\alpha}$. $\qquad\square$

## D.2 Proof of Theorem 1

*Proof.* Using Assumption 2, we have that

$$f(\mathbf{x}_{t+1}) - f(\mathbf{x}_t)$$
$$\leq \langle \nabla f(\mathbf{x}_t), \mathbf{x}_{t+1} - \mathbf{x}_t \rangle + \frac{L}{2}\|\mathbf{x}_{t+1} - \mathbf{x}_t\|^2$$
$$= -\eta_t \langle \nabla f(\mathbf{x}_t), \mathbf{g}_t(\mathbf{x}_{t+\frac{1}{2}}) \rangle + \frac{L\eta_t^2}{2}\|\mathbf{g}_t(\mathbf{x}_{t+\frac{1}{2}})\|^2$$
$$= -\eta_t \langle \nabla f(\mathbf{x}_t), \mathbf{g}_t(\mathbf{x}_{t+\frac{1}{2}}) - \nabla f(\mathbf{x}_t) + \nabla f(\mathbf{x}_t) \rangle + \frac{L\eta_t^2}{2}\|\mathbf{g}_t(\mathbf{x}_{t+\frac{1}{2}})\|^2$$
$$= -\eta_t \|\nabla f(\mathbf{x}_t)\|^2 - \eta_t \langle \nabla f(\mathbf{x}_t), \mathbf{g}_t(\mathbf{x}_{t+\frac{1}{2}}) - \nabla f(\mathbf{x}_t) \rangle + \frac{L\eta_t^2}{2}\|\mathbf{g}_t(\mathbf{x}_{t+\frac{1}{2}})\|^2.$$

Taking expectation, then plugging Lemmas 1 and 2 in, we have

$$\mathbb{E}\big[f(\mathbf{x}_{t+1}) - f(\mathbf{x}_t)\big] \leq -\left(\eta_t - \frac{3L\eta_t^2}{2}\right)\mathbb{E}\big[\|\nabla f(\mathbf{x}_t)\|^2\big] + \frac{L\rho^2}{2} + L^3\eta_t^2\rho^2 + L\eta_t^2\sigma^2.$$

As the parameter selection ensures that $\eta_t \equiv \eta = \frac{\eta_0}{\sqrt{T}} \leq \frac{2}{3L}$, it is possible to divide both sides with $\eta$ and rearrange the terms to arrive at

$$\left(1 - \frac{3L\eta}{2}\right)\mathbb{E}\big[\|\nabla f(\mathbf{x}_t)\|^2\big] \leq \frac{\mathbb{E}\big[f(\mathbf{x}_t) - f(\mathbf{x}_{t+1})\big]}{\eta} + \frac{L\rho^2}{2\eta} + L^3\eta\rho^2 + L\eta\sigma^2.$$

Summing over $t$, we have

$$\left(1 - \frac{3L\eta}{2}\right)\frac{1}{T}\sum_{t=0}^{T-1}\mathbb{E}\big[\|\nabla f(\mathbf{x}_t)\|^2\big] \leq \frac{\mathbb{E}\big[f(\mathbf{x}_0) - f(\mathbf{x}_T)\big]}{\eta T} + \frac{L\rho^2}{2\eta} + L^3\eta\rho^2 + L\eta\sigma^2$$
$$\overset{(a)}{\leq} \frac{f(\mathbf{x}_0) - f^*}{\eta T} + \frac{L\rho^2}{2\eta} + L^3\eta\rho^2 + L\eta\sigma^2$$
$$= \frac{f(\mathbf{x}_0) - f^*}{\eta_0\sqrt{T}} + \frac{L\rho_0^2}{2\eta_0\sqrt{T}} + \frac{L^3\eta_0\rho_0^2}{T^{3/2}} + \frac{L\eta_0\sigma^2}{\sqrt{T}}$$

where (a) uses Assumption 1, and the last equation is obtained by plugging in the value of $\rho$ and $\eta$. This completes the proof to the first part.

For the second part of this theorem, we have that

$$\mathbb{E}\big[\|\nabla f(\mathbf{x}_t + \boldsymbol{\epsilon}_t)\|^2\big] = \mathbb{E}\big[\|\nabla f(\mathbf{x}_t + \boldsymbol{\epsilon}_t) + \nabla f(\mathbf{x}_t) - \nabla f(\mathbf{x}_t)\|^2\big]$$
$$\leq 2\mathbb{E}\big[\|\nabla f(\mathbf{x}_t)\|^2\big] + 2\mathbb{E}\big[\|\nabla f(\mathbf{x}_t + \boldsymbol{\epsilon}_t) - \nabla f(\mathbf{x}_t)\|^2\big]$$
$$\leq 2\mathbb{E}\big[\|\nabla f(\mathbf{x}_t)\|^2\big] + 2L^2\rho^2$$
$$= 2\mathbb{E}\big[\|\nabla f(\mathbf{x}_t)\|^2\big] + \frac{2L^2\rho_0^2}{T}.$$

Averaging over $t$ completes the proof. □

## D.3 Proof of Theorem 2

*Proof.* To bound the MSE, we first have that

$$\|\mathbf{d}_t - \nabla f(\mathbf{x}_t)\|^2 \tag{9}$$
$$= \|(1-\theta)\mathbf{d}_{t-1} + \theta\mathbf{g}_t(\mathbf{x}_t) - (1-\theta)\nabla f(\mathbf{x}_t) - \theta\nabla f(\mathbf{x}_t)\|^2$$
$$= (1-\theta)^2\|\mathbf{d}_{t-1} - \nabla f(\mathbf{x}_t)\|^2 + \theta^2\|\mathbf{g}_t(\mathbf{x}_t) - \nabla f(\mathbf{x}_t)\|^2$$
$$\qquad + 2\theta(1-\theta)\langle \mathbf{d}_{t-1} - \nabla f(\mathbf{x}_t), \mathbf{g}_t(\mathbf{x}_t) - \nabla f(\mathbf{x}_t) \rangle.$$

Now we cope with three terms in the right hind of (9) separately.

The second term can be bounded directly using Assumption 2

$$\mathbb{E}\big[\|\mathbf{g}_t(\mathbf{x}_t) - \nabla f(\mathbf{x}_t)\|^2 | \mathbf{x}_t\big] \le \sigma^2. \tag{10}$$

For the third term, we have

$$\mathbb{E}\big[\langle \mathbf{d}_{t-1} - \nabla f(\mathbf{x}_t), \mathbf{g}_t(\mathbf{x}_t) - \nabla f(\mathbf{x}_t)\rangle | \mathbf{x}_t\big] = 0. \tag{11}$$

The first term is bounded through

$$\|\mathbf{d}_{t-1} - \nabla f(\mathbf{x}_t)\|^2 = \|\mathbf{d}_{t-1} - \nabla f(\mathbf{x}_{t-1}) + \nabla f(\mathbf{x}_{t-1}) - \nabla f(\mathbf{x}_t)\|^2$$

$$\overset{(a)}{\le} (1+\lambda)\|\mathbf{d}_{t-1} - \nabla f(\mathbf{x}_{t-1})\|^2 + \big(1 + \frac{1}{\lambda}\big)\|\nabla f(\mathbf{x}_{t-1}) - \nabla f(\mathbf{x}_t)\|^2$$

$$\le (1+\lambda)\|\mathbf{d}_{t-1} - \nabla f(\mathbf{x}_{t-1})\|^2 + \big(1 + \frac{1}{\lambda}\big)L^2\|\mathbf{x}_{t-1} - \mathbf{x}_t\|^2$$

$$= (1+\lambda)\|\mathbf{d}_{t-1} - \nabla f(\mathbf{x}_{t-1})\|^2 + \big(1 + \frac{1}{\lambda}\big)\eta^2 L^2\|\mathbf{g}_{t-1}(\mathbf{x}_{t-\frac{1}{2}})\|^2$$

where (a) is because of Young's inequality. Taking expectation and applying Lemma 2, we have that

$$\mathbb{E}\big[\|\mathbf{d}_{t-1} - \nabla f(\mathbf{x}_t)\|^2\big] \tag{12}$$

$$\le (1+\lambda)\mathbb{E}\big[\|\mathbf{d}_{t-1} - \nabla f(\mathbf{x}_{t-1})\|^2\big] + \big(1 + \frac{1}{\lambda}\big)\eta^2 L^2\Big(2L^2\rho^2 + 2\mathbb{E}\big[\|\nabla f(\mathbf{x}_{t-1})\|^2\big] + 2\sigma^2\Big)$$

$$\le (1+\lambda)\mathbb{E}\big[\|\mathbf{d}_{t-1} - \nabla f(\mathbf{x}_{t-1})\|^2\big] + \big(1 + \frac{1}{\lambda}\big) \cdot \mathcal{O}\Big(\frac{\sigma^2}{\sqrt{T}}\Big).$$

The last inequality uses the value of $\eta = \frac{\eta_0}{\sqrt{T}}$ and $\rho = \frac{\rho_0}{\sqrt{T}}$. In particular, we have $\eta^2\rho^2 L^4 = \mathcal{O}(1/T^2)$ and $\eta^2 L^2\sigma^2 = \mathcal{O}(\sigma^2/T)$, and

$$\eta^2 L^2 \mathbb{E}\big[\|\nabla f(\mathbf{x}_t)\|^2\big] = \frac{\eta_0^2 L^2}{T}\mathbb{E}\big[\|\nabla f(\mathbf{x}_t)\|^2\big] \le \eta_0^2 L^2 \frac{1}{T}\sum_{t=0}^{T-1}\mathbb{E}\big[\|\nabla f(\mathbf{x}_t)\|^2\big] = \mathcal{O}\Big(\frac{\sigma^2}{\sqrt{T}}\Big)$$

where the last equation is the result of Theorem 1.

Combining (9) with (12), (10) and (11), and choosing $\lambda = \frac{\theta}{1-\theta}$, we have

$$\mathbb{E}\big[\|\mathbf{d}_t - \nabla f(\mathbf{x}_t)\|^2\big] \le (1-\theta)\mathbb{E}\big[\|\mathbf{d}_{t-1} - \nabla f(\mathbf{x}_{t-1})\|^2\big] + \frac{(1-\theta)^2}{\theta}\mathcal{O}\Big(\frac{\sigma^2}{\sqrt{T}}\Big) + \theta^2\sigma^2$$

$$\le \theta\sigma^2 + \mathcal{O}\Big(\frac{(1-\theta)^2\sigma^2}{\theta^2\sqrt{T}}\Big)$$

where the last inequality is the result of Lemma 3. □

## D.4 Proof of Theorem 3

*Proof.* We adopt a unified notation for simplicity. Let $\mathbf{v}_t := \mathbf{d}_t$ for VaSSO, and $\mathbf{v}_t := \mathbf{g}_t(\mathbf{x}_t)$ for SAM. Then for both VaSSO and SAM, we have that

$$f(\mathbf{x}_t) + \langle \mathbf{v}_t, \boldsymbol{\epsilon}_t\rangle = f(\mathbf{x}_t) + \rho\|\mathbf{v}_t\| = f(\mathbf{x}_t) + \rho\|\mathbf{v}_t - \nabla f(\mathbf{x}_t) + \nabla f(\mathbf{x}_t)\|. \tag{13}$$

For convenience, let $\boldsymbol{\epsilon}_t^* = \rho\nabla f(\mathbf{x}_t)/\|\nabla f(\mathbf{x}_t)\|$. From (13), we have that

$$f(\mathbf{x}_t) + \langle \mathbf{v}_t, \boldsymbol{\epsilon}_t\rangle = f(\mathbf{x}_t) + \rho\|\mathbf{v}_t - \nabla f(\mathbf{x}_t) + \nabla f(\mathbf{x}_t)\| \tag{14}$$

$$\le f(\mathbf{x}_t) + \rho\|\nabla f(\mathbf{x}_t)\| + \rho\|\mathbf{v}_t - \nabla f(\mathbf{x}_t)\|$$

$$= f(\mathbf{x}_t) + \langle \nabla f(\mathbf{x}_t), \boldsymbol{\epsilon}_t^*\rangle + \rho\|\mathbf{v}_t - \nabla f(\mathbf{x}_t)\|.$$

Applying triangular inequality $\big|\|\mathbf{a}\| - \|\mathbf{b}\|\big| \le \|\mathbf{a} - \mathbf{b}\|$, we arrive at

$$f(\mathbf{x}_t) + \langle \mathbf{v}_t, \boldsymbol{\epsilon}_t\rangle = f(\mathbf{x}_t) + \rho\|\nabla f(\mathbf{x}_t) - (\nabla f(\mathbf{x}_t) - \mathbf{v}_t)\| \tag{15}$$

$$\ge f(\mathbf{x}_t) + \rho\|\nabla f(\mathbf{x}_t)\| - \rho\|\mathbf{v}_t - \nabla f(\mathbf{x}_t)\|$$

$$= f(\mathbf{x}_t) + \langle \nabla f(\mathbf{x}_t), \boldsymbol{\epsilon}_t^*\rangle - \rho\|\mathbf{v}_t - \nabla f(\mathbf{x}_t)\|.$$

Combining (14) with (15), we have

$$|\mathcal{L}_t(\mathbf{v}_t) - \mathcal{L}_t(\nabla f(\mathbf{x}_t))| \leq \rho\|\mathbf{v}_t - \nabla f(\mathbf{x}_t)\|$$

which further implies that

$$\mathbb{E}\big[|\mathcal{L}_t(\mathbf{v}_t) - \mathcal{L}_t(\nabla f(\mathbf{x}_t))|\big] \leq \rho\mathbb{E}\big[\|\mathbf{v}_t - \nabla f(\mathbf{x}_t)\|\big] \leq \rho\sqrt{\mathbb{E}\big[\|\mathbf{v}_t - \nabla f(\mathbf{x}_t)\|^2\big]}.$$

The last inequality is because $(\mathbb{E}[a])^2 \leq \mathbb{E}[a^2]$. This theorem can be proved by applying Assumption 3 for SAM and Lemma 2 for VaSSO. □

