# OpenReview forum: "Enhancing Sharpness-Aware Optimization Through Variance Suppression"
_NeurIPS.cc/2023/Conference — NeurIPS 2023 poster_

### Official Review · Reviewer_988S · 2023-06-27

**Soundness:** 2 fair
**Presentation:** 2 fair
**Contribution:** 3 good
**Rating:** 4
**Confidence:** 4

**Summary:**

This paper proposes a method, which applies EMA on batch gradients, to better solve the maximum problem in SAM and obtain higher test accuracy.

**Strengths:**

1.The proposed method is easy to implement.

2.The theoretical analysis is sufficient.

**Weaknesses:**

1.The core of this paper is to alleviate the gradient noise when solving the inner maximum problem of SAM. To achieve this goal, it proposes to implement EMA on the batch gradient and gives theoretical proof on the variance suppression. Although experiment results show the improvement of VaSSO, I am somehow confused that why the gradient noise harms the generalization performance of SAM. In the Section 4.1 of SAM[1], it claims that smaller batch size tends to yield models having better generalization ability, which conflicts the core idea of this paper.

2.Following 1, the statements on Line 47-48 and Line 50 need more support or evidence. And I think the introduction takes too much space to describe related works. In contrast, the author should give some experiment results to support the statement on Line 47-50.

[1] Foret P, Kleiner A, Mobahi H, et al. Sharpness-aware minimization for efficiently improving generalization[J]. arXiv preprint arXiv:2010.01412, 2020.

**Questions:**

no question

**Limitations:**

no limitation

---

> ### Author Rebuttal · Authors · 2023-08-08
>
> Thanks for reviewing this submission. Responses to the issues raised are provided next.
>
> **W1.** Most of existing works on m-sharpness only test SAM itself, but not SAM variants. Hence, m-sharpness does not necessarily generalize to our setting, simply because we introduce bias in $d_t$. Please see more experiments and explanation in the general response, where we numerically confirm that i) m-sharpness depends on the choice of neural network; ii) with the appearance of bias, m-sharpness even may not hold; and, iii) m-sharpness heavily depends on the specific means of updating SAM, which may not hold for different choices of $\epsilon_t$.
>
> **W2.**
> We will update our manuscript based on these responses.

---

> > ### Author Response · Authors · 2023-08-14
> > **Follow up**
> >
> > Dear reviewer 988S, we hope that our responses have addressed your concerns. Could you please let us know if further clarification is needed?

---

### Official Review · Reviewer_65jc · 2023-06-29

**Soundness:** 2 fair
**Presentation:** 3 good
**Contribution:** 2 fair
**Rating:** 5
**Confidence:** 4

**Summary:**

This paper proposes to improve upon sharpness-aware minimization (SAM) with variance-reduced inner perturbation steps. Theoretical analysis demonstrates that the proposed method achieves similar convergence rate with the original SAM. Empirical results on different applications (image classification and neural machine translation) demonstrate the effectiveness of proposed method. In addition, the proposed method also endows SAM with robustness against large label noise.

**Strengths:**

- The proposed method is clear and easy to understand
- Theoretical results are sound
- Empirical results demonstrate the superiority of proposed method


**Weaknesses:**

- Some claims are not fully supported
- Some baseline methods are missing


**Questions:**

- I am a bit puzzled by the claim that the proposed method can be easily integrated with computational efficient variants of SAM, e.g., (Liu et al, 2022; Zhao et al., 2022b) as well as a missing reference (Jiang et al., 2023). Compared with SAM that only needs to compute stochastic gradient, VaSSO has to keep track of d_t, as in (4a). I am not sure if the update is still possible if we only perform SAM step periodically (Liu et al, 2022) or randomly (Zhao et al., 2022b). The authors need to elaborate more on that.
- If such integration is not possible, the proposed method may suffer from larger computational cost than these variants. Although it can be regarded as the cost of better final performance, the authors may still need to make this limitation clear.
- Despite these computation efficient versions, the authors claim that some other existing works on SAM are also orthogonal to their work and can be easily integrated. While I do not find critical problems for these works, the authors may need to add some experiments that integrate the proposed method with these methods, and see if such integration can achieve any improvements. Nevertheless, such experiments are missing in current version.
- It also surprises me that the experiments contain so few baseline methods. For example, why is Fisher SAM (Kim et al., 2022) not compared in experiments? It is confusing to compare with ASAM and not Fisher SAM.

References:

Weisen Jiang, Hansi Yang, Yu Zhang, James Kwok. An adaptive policy to employ sharpness-aware minimization. ICLR 2023


**Limitations:**

Please see Weakness and Questions part.

---

> ### Author Rebuttal · Authors · 2023-08-08
>
> Thank you for the time spent for coping with our submission. We hope your concerns can be addressed after reading our responses.
>
> **Q1 & Q2.**  Since updating $d_t$ in (4a) only needs the gradient on the original model , i.e., $g_t(x_t)$, which is computed every iteration, our work can be adopted jointly with the computational efficient variants  (Liu et al, 2022; Zhao et al., 2022b). For example, we combine our VaSSO with (Zhao et al., 2022b) in the pseudocode below.
>
> - For t = 0 ... T
>      - Draw a Bernoulli random variable $B_t$ ($B_t = 0$ with probability p)
>      - Calculate g_t(x_t)
>      - Update $d_t = (1- \theta) d_{t-1} + \theta g_t(x_t)$
>      - If $B_t = 1$, update with SGD, i.e., $x_{t+1} = x_t - \eta_t  g_t(x_t)$
>      - If $B_t = 0$, update with VaSSO, i.e., $\epsilon_t = \rho d_t /|| d_t ||$ and $x_{t+1} = x_t - \eta_t  g_t(x_t + \epsilon_t)$
> - EndFor
>
> We also include some numerical results with $p=0.3$ on CIFAR10  to demonstrate the efficiency of VaSSO in this case.
>
> |                       |    (Zhao et al., 2022b)   |          VaSSO+ (Zhao et al., 2022b) |
> | -------------    | ---------------    | -------------
> | ResNet18      | 96.37  $\pm$ 0.13  | 96.50 $\pm$ 0.16|
>
>
> **Q3.** We test VaSSO aided ASAM on CIFAR10, and the results show that VaSSO helps ASAM.
>
> |                       |   ASAM   |          VaSSO+ASAM
> | -------------    | ---------------    | -------------
> | ResNet18      | 96.33 $\pm$ 0.09  | 96.52 $\pm$ 0.12|
> | WRN-28-10   | 97.15 $\pm$ 0.05  |  97.46 $\pm$ 0.08 |
>
> **Q4.** The numerical tests for FisherSAM on CIFAR10 are shown below. FisherSAM performs slightly worse than VaSSO.
>
> |                       |   FisherSAM  |          VaSSO |
> | -------------    | ---------------          | -------------
> | ResNet18      | 96.73 $\pm$ 0.03   | 96.77 $\pm$ 0.09|
> | WRN-28-10   | 97.46 $\pm$ 0.18   |  97.54 $\pm$ 0.12 |
> | PyramidNet110| 97.84 $\pm$ 0.11  |  97.93 $\pm$ 0.08|

---

> > ### Author Response · Authors · 2023-08-14
> > **Follow up**
> >
> > Dear reviewer 65jc, we hope that our responses have addressed your concerns. Could you please let us know if further clarification is needed?

---

> > > ### Comment · Reviewer_65jc · 2023-08-18
> > >
> > > I have checked your responses as well as comments from other reviewers. I think the additional experiments addressed most of my concerns, so I increased my score. The authors should add these additional experiments to the camera ready version.

---

> > > > ### Author Response · Authors · 2023-08-20
> > > >
> > > > We are glad to address your concerns. The discussions above will be included in the polished paper for sure.

---

### Official Review · Reviewer_BP9H · 2023-07-03

**Soundness:** 3 good
**Presentation:** 3 good
**Contribution:** 3 good
**Rating:** 5
**Confidence:** 3

**Summary:**

This paper proposes a variance suppression approach for SAM in order to account for the sensitivity of stochastic gradients used in SAM inner maximization. The proposed method is shown to provably reduce the MSE of gradient estimation. Some experimental results on benchmark datasets are provided.

**Strengths:**

Paper strengths:
1. great motivation for variance suppression
2. simple algorithm modification to standard SAM
3. theoretical results for VaSSO in Theorem 2 and Corollary 1

**Weaknesses:**

Paper weaknesses:
1. The authors claim in section 3.2 that VaSSO can boost performance of other SAM family approaches, but this is not shown in the experimental results
2. gains are marginal for larger datasets, see Table 3
3. authors provide convergence rates for Vasso in Corollary 1 but it's not clear that these are sharper than the SAM rates in Theorem 1



**Questions:**

Questions to address in rebuttal:
1. does VaSSO translate to gains in domain generalization performance? e.g. WILDS benchmark [1]
2. can authors cite more Frank-Wolfe papers in connection to min-max problems, e.g. [2], [3]
3. how do convergence rates in Corollary 1 relate to Frank-Wolfe optimization convergence rates? convergence rates for FW algorithms and various variants have been studied in the literature [4]
4. what are the limitations of VaSSO? authors should include a discussion of limitations
5. can the authors add more SAM baselines to the list? Also, ASAM combined with VaSSO is not included, and would be good to include for comparisons

I am willing to increase my score if the authors address most of these concerns.

[1] Koh et al, WILDS: A Benchmark of in-the-Wild Distribution Shifts, https://arxiv.org/abs/2012.07421

[2] Tsiligkaridis et al, Understanding and Increasing Efficiency of Frank-Wolfe Adversarial Training, CVPR 2022, https://arxiv.org/abs/2012.12368

[3] Gidel et al, Frank-Wolfe Algorithms for Saddle Point Problems, https://arxiv.org/pdf/1610.07797.pdf

[4] Huang et al, Accelerated Stochastic Gradient-free and Projection-free Methods, ICML 2020


**Limitations:**

Not discussed.

---

> ### Author Rebuttal · Authors · 2023-08-08
>
> We thank the reviewer for the time devoted for this work. We find the comments helpful to improve the quality of our work, and we are happy to modify our manuscript accordingly.
>
> **W1.**  We have combined VaSSO with ASAM, and here are our results for CIFAR10. It can be seen that VaSSO+ASAM outperforms ASAM.
>
> |                       |   ASAM   |          VaSSO+ASAM
> | -------------    | ---------------    | -------------
> | ResNet18      | 96.33 $\pm$ 0.09  | 96.52 $\pm$ 0.12|
> | WRN-28-10   | 97.15 $\pm$ 0.05  |  97.46 $\pm$ 0.08 |
>
> **W2.** We respectfully disagree with this assessment. The accuracy is not easy to improve when using ResNet50 on the ImageNet. In addition, SAM improves over SGD by 0.54, and VaSSO further improves over SAM by 0.28. This is a 52% extra improvement compared to SAM’s merit over SGD, which is not marginal.
>
> **W3.** The convergence rates are the same.
>
> **Q1.** Thanks for pointing out another possible domain for VaSSO. While domain adaptation is not our central theme, it will be of interest to investigate a possible extension of VaSSO to this end. In fact, we have noticed that other sharpness aware optimization approaches have been applied to domain adaptation; see e.g., [5].  We have this direction in our future research agenda, and look forward to leverage the problem structure of domain adaption for further improvement on top of VaSSO.
>
> [5] P. Wang, et al. "Sharpness-aware gradient matching for domain generalization." In Proceedings of the IEEE/CVF Conference on Computer Vision and Pattern Recognition, pp. 3769-3778. 2023.
>
>
> **Q2.** Thanks for pointing out missing references. We will include them in the updated manuscript.
>
> **Q3.** Our rate is not directly comparable to [4]. This is because the inner maximization of SAM/VaSSO is changing over iterations, i.e., $\max_{||\epsilon|| \leq \rho}f(x_t + \epsilon)$; and SAM/VaSSO only uses 1-step FW for solving its inner maximization problem per iteration; see more in Appendix 1. These differences make it difficult to compare our rates with [4]. However, it may be possible to apply methods in [4] to our framework for better bounds. We will discuss these issues when outlining our future work in the revised paper.
>
> **Q4.** The limitation of VaSSO is that it has to compute gradient twice per iteration, which we have discussed in line 329 - 331. We have also included potential solutions in the same paragraph, with numerical results shown in our responses to Q1 and Q2 of Reviewer 65jc.
>
> **Q5.** The combination of VaSSO and ASAM is shown earlier in our response to W1. For other SAM baselines, we adopt FisherSAM (Kim et al, 2022). The test accuracy on CIFAR10 is shown below, where VaSSO also has numerical merits over FisherSAM
>
>
> |                       |   FisherSAM  |          VaSSO |
> | -------------    | ---------------          | -------------
> | ResNet18      | 96.72 $\pm$ 0.03   | 96.77 $\pm$ 0.09|
> | WRN-28-10   | 97.46 $\pm$ 0.18   |  97.54 $\pm$ 0.12 |
> | PyramidNet110| 97.84 $\pm$ 0.11  |  97.93 $\pm$ 0.08|
>
>
> We hope our responses address your concern. Let us know if there are further comments and we are happy to discuss.

---

> > ### Author Response · Authors · 2023-08-14
> > **Follow up**
> >
> > Dear reviewer BP9H, we hope that our responses have addressed your concerns. Could you please let us know if further clarification is needed?

---

> > > ### Comment · Reviewer_BP9H · 2023-08-14
> > >
> > > I thank the authors for their responses. I have decided to increase my score.

---

### Official Review · Reviewer_zZon · 2023-07-11

**Soundness:** 2 fair
**Presentation:** 3 good
**Contribution:** 2 fair
**Rating:** 4
**Confidence:** 4

**Summary:**

**I have read the author's rebuttal, see reply below** This work proposes changing SAM from perturbing the weights by the gradient of the current minibatch, with instead a moving average of the gradients across training iterations. They provide upper bounds that indicate that the moving average may better approximates the inner-maximization with respect to the loss over the entire dataset, which is more difficult for naive SAM due to the minibatch noise. Empirically, SAM with moving average gradients (which they call VaSSO) performs better on CIFAR10, CIFAR100 by around $0.1-0.5$%, and the improvements are more apparent in the presence of heavy label noise with gains up to +10% improvement with 75% label noise. Empirical experiments show that VaSSO minimizes the maximum eigenvalue of the Hessian better than SAM.

**Strengths:**

**Significance**: The proposed algorithm VaSSO is a very simple change that achieves flatter minima and achieves better generalization than SAM. Although it may not necessarily be the case that two effects are tied in a causal manner, its effectiveness at achieving both beyond SAM may suggest that it may be a useful optimizer in practice and for further study in future works.

**Quality**: The paper is clearly written and organized.

**Weaknesses:**

**1. Lack of important literature review**: The authors motivate VaSSO by identifying problems with SAM in terms of its ability to optimize the original objective _due to the minibatch noise_. The moving average is hypothesized to be better at correctly approximating the full-batch gradient. The authors do not address the fact that works that have analyzed SAM previously have unanimously observed that SAM actually **only observes improvements in generalization if it is paired with minibatch noise**. In particular, Andruschenko et al (https://arxiv.org/abs/2206.06232) showed that n-SAM which utilizes the full-batch gradient directly for the perturbation step actually observed no generalization gain. This is supported by experiments on m-sharpness in the original SAM paper (https://arxiv.org/abs/2010.01412), and they also showed that better approximating the sharpest direction by also taking the second order approximation of the loss instead of the first order approximation lead to worse performance. Wen et al (https://arxiv.org/abs/2211.05729) showed that the minibatch noise is important to minimize the trace of the Hessian instead of the max eigenvalue (though the work does not make any claim about which measure of sharpness is better correlated with generalization).

To summarize, the authors pose the problem as that SAM cannot achieve the full generalization benefits because it does not minimize its intended objective optimally, but previous literature indicate that this suboptimality is actually what allows SAM to achieve better generalization. The authors do not mention the conclusions made in previous works, but it seems important to address this conflict.

**2. Marginal improvements, and inconsistent numbers in comparison to Foret et al.**: Without label noise VaSSO's improvements range between 0.2-0.5% for CIFAR10 and CIFAR100, and it's not clear whether further hyperparameter tuning could close this gap. In particular, the authors train WideResNet-28-10 on CIFAR10 with cutout data augmentation which was also conducted by Foret et al, where they report an error of 2.3% while the authors report 2.7% and for VaSSO 2.5%. This slight improvement may potentially come from the authors only optimizing the rho hyperparameter, and not m-sharpness (the other hyperparameter mentioned in Foret et al. where the minibatch is sharded first).

For label noise, there is a much more significant boost in test accuracy, but it is unclear whether the authors are reporting the peak early-stopping accuracy or the final accuracy. The former is more important for label noise, and the difference between VaSSO and SAM may be coming from reporting the latter. Also, Foret et al reported better numbers for SAM on CIFAR10 with similar amounts of label noise although they were using ResNet34 instead of ResNet18.

**3. The upper bounds are too loose, and comparing upper bounds does not lead to any meaningful comparison.** The upper bound derived in Theorem 2 requires very approximate intermediate steps (very loose upper bound in Line 181), and is not a function of $t$ but $T$. The bound implies that if you train for longer, the bound for the MSE between the moving average and the full batch gradient _at every intermediate step_ improves which doesn't seem right.

More importantly, any meaningful comparison between the MSE for SAM and MSE for VaSSO should compare the upper bound of VaSSO to the _lower bound_ for SAM. Comparing two upper bounds isn't very meaningful.

**Questions:**

See weaknesses

---

> ### Author Rebuttal · Authors · 2023-08-08
>
> Thank you for the time devoted to this review. The issues raised are addressed one-by –one next.
>
> **W1.** It is prudent to clarify that *m-sharpness is only tested for SAM, which may not necessarily generalize to other SAM variants*. One of the key differences with m-sharpness in SAM is that VaSSO introduces **bias** in $d_t$. As shown in our **general response,**  VaSSO works in a regime where m-sharpness does not necessarily hold. Thus, the role of noise in SAM does not imply that for VaSSO.
>
> In addition to that, there are some other concerns that we hope to settle in our response.
>
> >  Andruschenko et al .. n-SAM .. actually observed no gain
>
> - VaSSO does not reduce to SAM even when using full gradient when finding $\epsilon_t$; hence, the results of the aforementioned citation may not generalize to VaSSO. Moreover, n-SAM does not account for the normalization step in SAM. This can make a great difference especially when the stochastic gradient noise is pronounced. In addition, the implementation in this citation does not fully match the claim, as n-SAM is implemented by performing the `ascent step on a different batch compared to the descent step,' as shown in Appendix D of the aforementioned citation.
>
> - The experiment in this citation only observes that SAM has an accuracy of 95.8 using ResNet18 on CIFAR10, which is lower than our implementation of SGD (96.25) or SAM (96.58). This is perhaps  because a piecewise linear learning rate is adopted in this citation or no data augmentation (such as cutout) has been applied. Therefore, the results in this citation may not necessarily generalize to our setting since the *cutout* might have markedly changed the loss curvature.
>
> >  pose the problem as that SAM cannot achieve the full benefits because it does not minimize its intended objective optimally
>
> There may have been a misunderstanding. We do not claim that solving the inner maximization optimally is helpful. We only point out that the approximation in the current SAM derivation is not perfect. And this is a chance for making SAM stronger. In fact, even if the inner maximization uses the full gradient, it is unlikely to solve the inner maximization optimally in a single step.
>
> **W2.**
>
> > Marginal improvements
>
> We will respectfully disagree with this assessment. Most of our numerical results show a clear improvement over SAM, especially those in  Tables 3 and 6.
>
> > inconsistent numbers in comparison to Foret et al
>
> There are several reasons for the difference on WRN. The first is that [Foret et al] uses JAX, while we work with pytorch. The second reason is that  [Foret et al] works with 8 GPUs, which allows calculation of 8 adversarial models $ \epsilon_t^i$ for $i \in 1, 2, .., 8$ and computes $g_t^i(x_t + \epsilon_t^i)$ separately on each GPU. Unfortunately, this approach requires each GPU to backpropagate twice (16 times in the 8-GPU setup), which is unaffordable by our single GPU setting.
>
> > peak early-stopping accuracy or the final accuracy
>
> As suggested, we report the peak early stopping accuracy when the noise level is 75%. SAM achieves 76.02, while VaSSO exhibits an accuracy of 83.63, which is still a considerable improvement.
>
> >  Foret et al reported better numbers for SAM on CIFAR10 although they were using ResNet34 instead of ResNet18.
>
> Our difference is that [Foret et al] trains a **ResNet32,** which has only 0.46M parameters (there is a **typo** in the review, not ResNet34). We are training a ResNet18, which is an 11M model. Our model is more than 20x larger than that in [Foret et al]; hence, the difference is reasonable.
>
> **W3.**
>
> > Theorem 2 dependent on T but not t.
>
> This T dependence comes from the choice of $\rho = O(1/\sqrt{T})$ and $\eta = O(1/\sqrt{T})$. This is mainly to simplify the proof. However, extending the choice of hyperparameters to $\rho_t = O(1/\sqrt{t})$ and $\eta_t = O(1/\sqrt{t})$ is rather straightforward based on standard optimization techniques. For such a parameter choice, Theorem 2 can be readily modified to exhibit dependence of order $O(1 / \sqrt{t} )$.
>
> > Comparing upper bounds does not lead to any meaningful comparison
>
> Unfortunately, the reviewer *missed the fact that VaSSO is compared with a lower bound*.
>
> Recall that a lower bound means that there exists an instance, such that $\mathbb{E}[ || g_t(x_t) - \nabla f(x_t) ||^2]= \sigma^2$. The simple 1d example provided next shows that this is indeed a lower bound.
>
> Let $f(x, \xi) = h(x) + \xi x$, where h(x) is a deterministic loss function, and $\xi$ is a Gaussian random variable with 0 mean and variance $\sigma^2$. For such a loss function, it can be readily verified that
> $\mathbb{E}[| | g_t(x_t) - \nabla f(x_t) ||^2] = \sigma^2$. Hence, our bound on SAM is indeed a lower bound, and the comparison is certainly meaningful.

---

> > ### Author Response · Authors · 2023-08-14
> > **Follow up**
> >
> > Dear reviewer zZon, we hope that our responses have addressed your concerns. Could you please let us know if further clarification is needed?

---

### Author Rebuttal · Authors · 2023-08-08

**General response to Reviewers zZon and 988S**: Our results do not conflict with existing works.

Specifically, m-sharpness does not conflict with the contribution of this submission for three reasons:

- the behavior of m-sharpness may depend on the adopted dataset and neural network; see Experiment 1.
- m-sharpness may not hold when the gradient estimator is biased, which is the case of VaSSO; see Experiment 2.
- m-sharpness depends strongly on the specific SAM update, and its generalizability to other approaches is still well supported; see Experiment 2.

Reasons above explain why our results do not conflict with m-sharpness, simply because the present submission deals with a regime that m-sharpness may not hold. We confirm this with the following experiments.

**Experiment 1**: m-sharpness on transformers

We conduct experiments on a transformer following the settings in section 4.2. We use a fixed batch size of B and vary the choices of m in {B/2, B/4}. For each m, we tune $\rho$ from {0.025, 0.05, 0.1, 0.2, 0.5}, respectively. The best BLEU score are reported below (larger is better).

|         |   SAM    |    B/2-SAM  |   B/4-SAM   |
| ---    | ---         | ---          | ---          |
| BLEU  | 34.75 $\pm$ 0.04  | 34.73 $\pm$ 0.02 |  34.69 $\pm$ 0.04  |

It can be seen that a smaller m actually worsens the test performance, suggesting that m-sharpness depends on the adopted neural networks and datasets.

**Experiment 2**: m-sharpness under the appearance of bias

A critical difference between m-sharpness in SAM and VaSSO is that the latter uses a biased gradient estimator $d_t$ to find $\epsilon_t$. To test the impact of bias in m-sharpness, consider the following experiment. First, fix a bias vector $\beta$. $\beta$ with the same size of gradients, and with each entry sampled from a distribution of ${\cal N}(0.01,1)$. Next, normalize $\beta$ to ensure $||\beta|| = 0.1$. We use $\beta$ as the bias to find the adversarial model, i.e., $\epsilon_t = \rho (g_t(x_t) + \beta)/ || g_t(x_t) + \beta ||$. We find that when bias is present, the m-sharpness may not hold true for ResNet18 on CIFAR10. Moreover, a decreasing trend on test accuracy is observed in this case, which illustrates how helpful is to reduce the gradient variance.

|     |   m=128    |    m=64  |   m=32   | m=16 |
| ---  | ---    | ---   | --- |  --- |
| BLEU  | 96.33 $\pm$ 0.04  | 96.26 $\pm$ 0.07 |  96.26 $\pm$ 0.10  |  96.18 $\pm$ 0.13|

Experiment 2 demonstrates also m-sharpness is highly related to the specific updateSAM is updated. Changing the way one finds $\epsilon_t$ can reverse the m-sharpness.

In sum, our Experiment 1 shows that m-sharpness depends on the neural network SAM is applied to. The exact reasoning behind m-sharpness is unclear, and it is clearly possible that m-sharpness is caused by a careless optimization step. In fact, most of the existing works on m-sharpness only test SAM, but do not consider alternative choices for inner maximization.

Lastly, it is important to stress the reason for not studying m-sharpness directly. We find that m-sharpness formulation, e.g., eq. (3) in Andruschenko et al 2022 *may be ill-posed mathematically due to the lack of a clear definition on how the dataset ${\cal S}$ is partitioned*. Using their notation,  suppose for instance that the loss function is $l_i(x) = a_i x^2 + b_i x $, where $(a_i, b_i)$ are data points. Consider a dataset with 4 samples, $(a_1=0, b_1=1)$; $(a_2=0, b_2=-1)$; $(a_3=-1, b_3=0)$; and, $(a_4=1, b_4=0)$. Let us consider 2-sharpness below under different partitions of the dataset.

- If the data partition is (1,2) and (3, 4), the objective of 2-sharpness i.e., (3) in Andruschenko et al 2022, becomes $\min_w \sum \max_{||\delta|| < \rho} 0$.
- If the data partition is (1,3) and (2,4), the objective is $\min_w \sum_{i=1}^2 \max_{||\delta|| < \rho} f_i(w,\delta)$, where $f_1$ is the loss on partition (1,3), i.e., $f_1(w,\delta) = -(w+\delta)^2 + (w+\delta)$; and $f_2(w,\delta) = (w + \delta)^2 - (w + \delta)$ is the loss on partition (2,4).

Clearly, the loss functions are not the same when the data partition varies.

---

### Decision · Program_Chairs · 2023-09-21

**Decision:**

Accept (poster)

**Comment:**

This paper studies SAM an points out a potential issue with the original algorithm which can adversely affect its behavior. SAM attempts to solve a minimax optimization, where the inner optimization is solved over a mini-batch. The paper points out that, the behavior of SAM can be very brittle due to sensitivity of the inner optimization to the draw of the mini-batch; one sample can drastically change the max output if the mini-batch is very small. To stabilize the inner optimization, the paper proposes a variance reduction scheme based on exponential moving average.

The paper received 4 reviews, and after the rebuttal two reviewers increased their initial score, and the paper ended up being a borderline score-wise. The main concern from the reviewers who maintained their initial scores, reviewers 988S and zZon, is some conflicting evidence in the literature that having a smaller mini-batch actually can help with generalization of SAM, an observation that is referred to as M-Sharpness. Authors' response to this concern is that, while there is some evidence about the role of M-Sharpness in generalization, the exact connection is still far from understood, and the effect of M-Sharpness on generalization can depend on add the choice of the dataset and architecture. This is backed up by a toy example in their general rebuttal response where they authors show how the different partition of the data can lead to suboptimal effect of M-Sharpness on generalization. I tend to agree with this response, and I find the main line of arguments in the paper (around the brittleness of the inner maximization) reasonable. The proposes scheme shows one cheap way to stabilize the inner optimization, and the empirical evidence in the paper suggests that this can indeed help in certain scenarios.

I encourage authors to incorporate the M-Sharpness arguments they presented in the rebuttal into the final version of the paper. Also, regarding related works on convergence of SAM, I encourage the authors to consider including a brief discussion about "Bouncing Across Ravines and Drifting Towards Wide Minima" by Bartlett et al.